# Synchrony of Bird Migration with Global Dispersal of Avian Influenza Reveals Exposed Bird Orders

Qiqi Yang [1] ✉, Ben Wang[2], Phillipe Lemey [3], Lu Dong[4], Tong Mu [5],
R. Alex Wiebe[1], Fengyi Guo [1], Nídia Sequeira Trovão [6], Sang Woo Park [1],
Nicola Lewis[7,8], Joseph L.-H. Tsui[9], Sumali Bajaj[9], Yachang Cheng[10], Luojun Yang[11],
Yuki Haba[1], Bingying Li[2], Guogang Zhang[12,13], Oliver G. Pybus[8,9,13],
Huaiyu Tian [2,13] ✉ & Bryan Grenfell [1,5,13] ✉

Highly pathogenic avian influenza virus (HPAIV) A H5, particularly clade 2.3.4.4, has caused worldwide outbreaks in domestic poultry, occasional spillover to humans, and increasing deaths of diverse species of wild birds since 2014. Wild bird migration is currently acknowledged as an important ecological process contributing to the global dispersal of HPAIV H5. However, this mechanism has not been quantified using bird movement data from different species, and the timing and location of exposure of different species is unclear. We sought to explore these questions through phylodynamic analyses based on empirical data of bird movement tracking and virus genome sequences of clade 2.3.4.4 and 2.3.2.1. First, we demonstrate that seasonal bird migration can explain salient features of the global dispersal of clade 2.3.4.4. Second, we detect synchrony between the seasonality of bird annual cycle phases and virus lineage movements. We reveal the differing exposed bird orders at geographical origins and destinations of HPAIV H5 clade 2.3.4.4 lineage movements, including relatively under-discussed orders. Our study provides a phylodynamic framework that links the bird movement ecology and genomic epidemiology of avian influenza; it highlights the importance of integrating bird behavior and life history in avian influenza studies.

Since 2014, highly pathogenic avian influenza viruses (HPAIVs) subtype H5, especially clade 2.3.4.4, have caused unprecedentedly large numbers of wild bird deaths worldwide[1]. In contrast to previous clades of the A/goose/Guangdong/1996 (Gs/GD) lineage, there have also been more persistent spillovers to local domestic poultry[2–6], impacting the poultry farming industry. Despite no onward human-to-human transmission to date, occurrences of zoonotic jumps to humans pose potential threats to public health[7–12].

Plausible ecological mechanisms for the global movement of HPAIVs include the live poultry trade and wild bird migration[13,14]. Endemic in Asia, clade 2.3.2.1 spread throughout the Eurasian and African continents (Fig. S2). Since its outbreak in wild birds at Qinghai Lake in 2005[15], studies have shown that viral dispersal at long distances is related to wild bird migration[16,17]. Global dispersal of clade 2.3.4.4 continues to provide virological, epidemiological, and ecological evidence in support of the critical role of migratory wild birds in HPAIV spread and evolution at a global-scale. Compared to previous clades, clade 2.3.4.4a during 2014/15 outbreaks was less pathogenic to some species while being more effectively transmitted[18–20], possibly enabling infected birds to migrate between continents. Subsequent phylodynamic work confirmed that the introduction of clade 2.3.4.4a into Europe and North America most likely occurred via long-distance

flights of infected migratory birds[21]. During the 2016/17 outbreaks, the major circulating clade 2.3.4.4b was more transmissible[22] and more virulent[23], related to multiple internal genes[22,23] and potentially their frequent reassortments[3–5]. Later phylogenetic analysis showed a clear link between the reassortments and migratory birds, as most reassorted gene segments were from migratory wild birds and originated at dates and locations that corresponded to their hosts' migratory cycles[24]. Integrating host movement in studying HPAIV dispersal is important, though challenging. One challenge is insufficient bird movement data, which limited previous global-scale studies[21] in accounting for the high variation in bird behaviors across species and locations.

Another challenge of studying HPAIV dispersal in wild birds is the lack of HPAIV prevalence data. Only a few studies document longitudinal HPAIV prevalence in wild bird populations[25]. Compared to HPAIV, low pathogenic AIV (LPAIV) has better long-term surveillance of infections or seroprevalence and related avian host ecology in disparate bird habitats, e.g., the United States Geological Survey surveillance of birds in Alaska[26]. While longitudinal records provide insights into the role of life history and ecology of local bird communities in LPAIV circulation[27], their conclusions are limited to local dynamics and cannot be easily generalized. To resolve this challenge, ideally, we should have systematic global surveillance for HPAIV. However, this is impossible due to resource constraints.

Instead, we could design effective surveillance strategies by identifying exposed avian species and high-risk geographical regions. Recently, researchers have addressed these questions at a higher taxonomic level to include more diverse species. For example, Hill et al. compared the different roles of species within the Anseriformes and Charadriiformes in the dispersal and spillover of AIVs[28]. They concluded that wild geese and swans are the main source species of HPAIV H5, while gulls spread the viruses most rapidly. Hicks et al. found that the inter-species transmission of AIVs in North America is positively associated with the overlap of habitats, suggesting the importance of local bird community diversity[29]. However, they did not use empirical bird movement data. Furthermore, given the heterogeneous biogeographical pattern of bird migrations, identifying geographical hotspots requires linking global and local scales.

To fill this gap, we here focus on two questions related to the contributions of birds, locally and globally, to the spatiotemporal dynamics of HPAIV H5 viruses; specifically, i) how does seasonal bird migration facilitate global virus dispersal and ii) which avian species are exposed to HPAIV H5 and where? To explore these questions, we first illustrate the global circulation history of clade 2.3.4.4 (2010–2017; 2018–2023) and clade 2.3.2.1 (2010–2017) using time-scaled phylogeographic analyses of hemagglutinin (HA) genes of HPAIVs sampled from wild birds and poultry. The split of clade 2.3.4.4 data is based on its history of evolution, epidemiology and sampling intensity, enabling us to maintain the genetic diversity of the dataset while keeping the computational burden manageable. Building upon previous evidence, we propose possible routes of long-distance virus dispersal. We acknowledge two caveats. First, while we only included HA, the neuraminidase (NA) and internal genes contribute to virus evolution, e.g., via reassortment[24] and the major circulating subtype shift from H5N8 to H5N1 in clade 2.3.4.4. Second, the geographical bias of virus sampling has a strong impact on the virus lineage movement routes, especially for locations under-sampled. Based on the estimated routes and inferred virus dispersal history, we quantify the contribution of seasonal bird migrations to global virus dispersal and evolution. Second, we model the monthly geographical distribution of bird orders using species distribution models based on environmental factors and bird tracking data. We evaluate the risks of bird orders being exposed to HPAIV H5 at geographical origins and destinations of virus lineage movement by analyzing the statistical association of local bird distributions and virus lineage migration. Our study provides an approach that integrates bird migration ecology in HPAIV epidemiological studies to disentangle the mechanisms of interaction between HPAIV and wild birds.

## Results
### Seasonal bird migration associates with global HPAIV H5 dispersal
Is the wide geographical range of HPAIV H5 caused by frequent introductions from one region to another, or a single introduction resulting in subsequent spread within the area? The discrete trait phylogeographical analysis of clade 2.3.4.4 HA genes (before 2018) exhibits scarce virus lineage movements between aggregated regions, most of which are unidirectional (Fig. 1B). It suggests that inter-regional viral introductions over long geographical distance occur at low frequency and in one direction. Furthermore, the sequences are highly clustered by region, implying viral persistence within each region after introduction. Clade 2.3.2.1 shows a similar pattern (Fig. S8). In contrast, clade 2.3.4.4 after 2018 (Fig. 1C) presents more mixed virus lineage movements between regions, suggesting more frequent dispersal between regions. In addition, the trunk composition of the phylogeny shows that most virus diversity is contained in Russia, Africa and Europe. While the two trees of clade 2.3.4.4 (before and after 2018) have shared branches (Fig. 1A), the resulting correlation does not qualitatively affect the subsequent analysis (Fig. 2, Fig. S4).

To test quantitatively whether seasonal bird migration is a key predictor of HPAIV H5 dispersal, we fit a generalized linear model (GLM) parameterization of the discrete phylogeography using a Bayesian model selection procedure[30,31]. Concurrently, we consider seasonal bird migration, live poultry trade and poultry population size as covariates of the diffusion rates between regions. To incorporate the potential seasonal difference in viral dispersal, we model a time-heterogeneous phylogenetic history[32] with three seasons based on bird annual cycle in North Hemisphere: non-migration (mid-November to mid-February, mid-May to mid-September), spring migration (mid-Feburary to mid-May) and fall migration (mid-September to mid-November). Fig. 2 shows the posterior estimates of the inclusion probabilities and conditional effect sizes (on a log scale) of the covariates. It reveals that seasonal bird migration is the dominant driver of the global virus lineage movements of HPAIV H5 clade 2.3.4.4. This is shown by both the log conditional effect size of the seasonal bird migration (for clade 2.3.4.4 (2010–2017), mean log conditional effect size [95% highest probability density interval, HPDI]:-2.35 [0.87, 3.85]; clade 2.3.4.4 (2018–2023):-4.71 [3.60, 6.22]) and the statistical support for its inclusion (for clade 2.3.4.4 (2010–2017), posterior probability (PP) ~ 0.88 and Bayes factor (BF) ~ 56; clade 2.3.4.4 (2018–2023), PP ~ 1 and BF > 16565).

In contrast, poultry population size and the international live poultry trade are not strongly or not consistently associated with the inter-region dispersal of HPAIV H5 (Fig. 2). This is also evident in both the effect size and the statistical support, e.g., the log conditional effect size of the live poultry trade (for clade 2.3.4.4 (2010–2017):-0.62 [0.19, 1.17]; clade 2.3.4.4 (2018–2023):-0.35 [0.08, 0.66]) and the statistical support for its inclusion (for clade 2.3.4.4 (2010–2017), PP ~ 0.60 and BF ~ 12; clade 2.3.4.4 (2018–2023), PP ~ 0.11 and BF ~ 1). For clade 2.3.2.1, neither wild bird migration nor live poultry trade is a major contributing predictor for the inter-regional virus lineage movements.

The viral sample size at origin location might affect the results (clade 2.3.4.4 (2010–2017):-−1.22 [−1.75, −0.68], PP ~ 1 and BF > 2647; clade 2.3.4.4 (2018–2023):-−0.86 [−1.13, −0.05], PP ~ 0.99 and BF ~ 866; clade 2.3.2.1: ~ −1.52 [−1.85, −1.19]), PP ~ 1 and BF > 16,565. Unfortunately, because of the imprecision of host species records, it is challenging to down-sample the data while maintaining genetic diversity. Nevertheless, when controlling for viral sample size at the origin

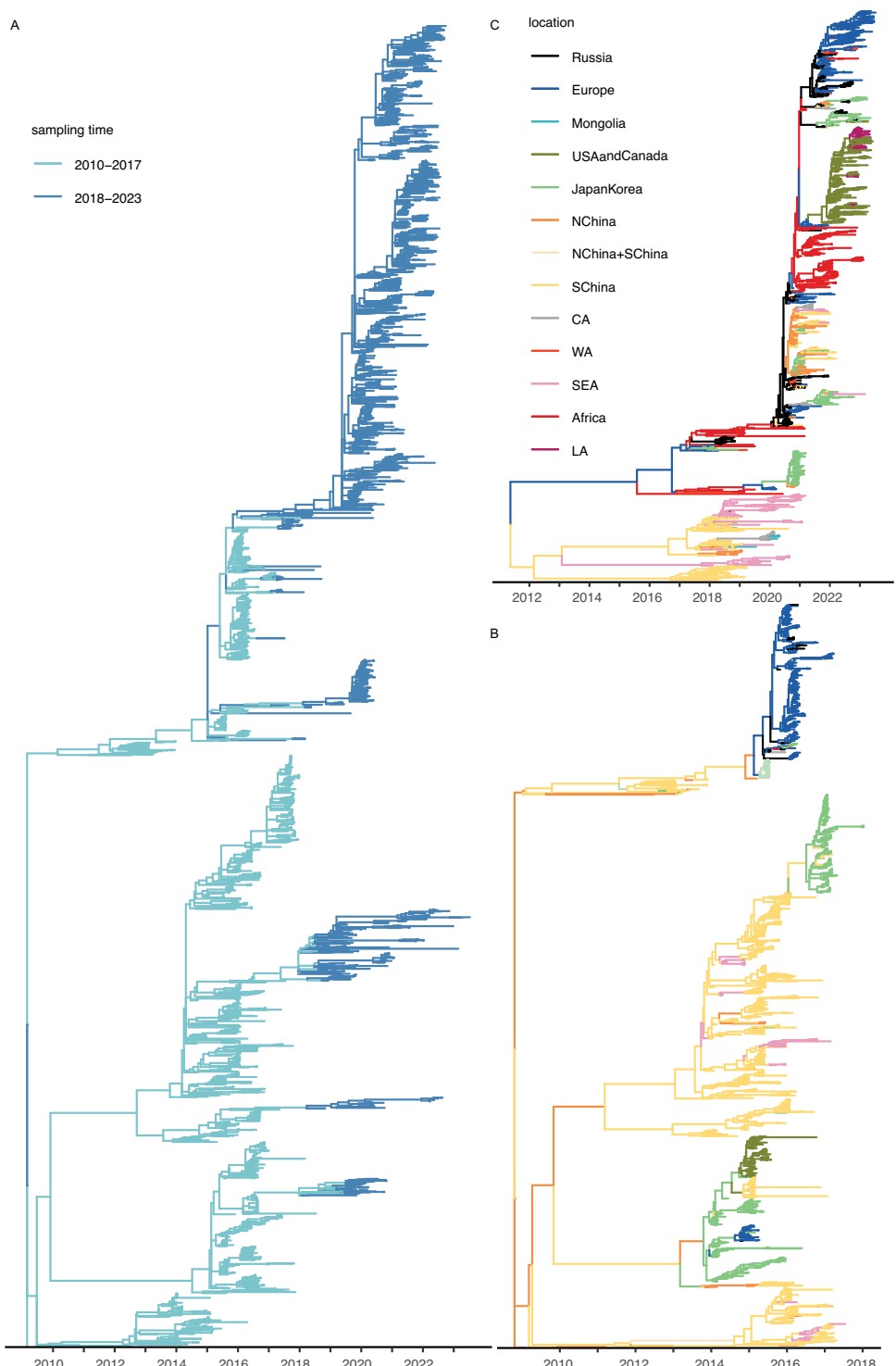

**Fig. 1 | The evolutionary history of HPAIV H5 clade 2.3.4.4 and 2.3.2.1. A** Time-scaled maximum clade credibility (MCC) tree of clade 2.3.4.4 (2010–2023). Sequences sampled from 2010 to 2017 are colored light blue, and those sampled from 2018 to 2023 are darker blue. The phylogeographic reconstructions for the sequences sampled from 2010 to 2017 (**B**) and for those from 2018 to 2023 (**C**), shown as time-scaled MCC trees with location annotations to summarize the reconstructions. Abbreviations of locations: NChina (North China), SChina (South China), SEA (South-East Asia), CA (Central Asia), WA (Western Asia), LA (Latin America). Supplementary Dataset 3 lists countries in each aggregated region.

location and directly comparing wild bird migration and live poultry trade, the wild bird migration is clearly supported over the live poultry trade (Table S2). Based on these results, we used subsequent analyses to understand the importance of different bird species at the order taxonomy level in the global dispersal and local emergence of HPAIV H5.

**Exposed migratory bird orders at origin and destination regions of HPAIV H5 virus lineage movement**

We identified virus dispersal routes (Bayes factor > 3) between the aggregated regions in the Northern Hemisphere (Fig. 3A) using the previous phylogeography analyses. Seasonality is reflected in northward and southward virus lineage movements of clade 2.3.4.4.

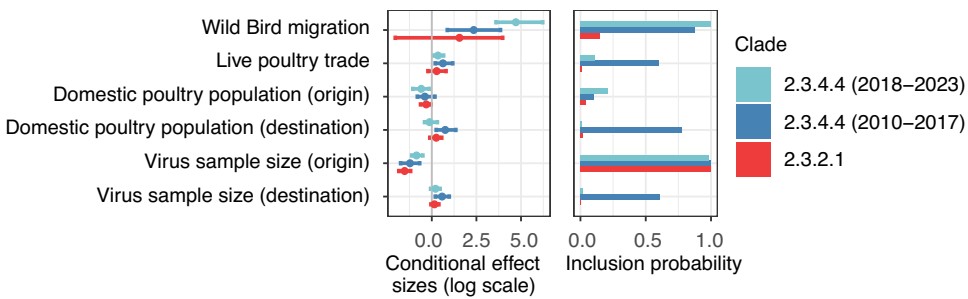

**Fig. 2 | Contributions of predictors to worldwide diffusion of H5N1 clade 2.3.2.1 and 2.3.4.4.** The virus dispersal is inferred from HA genes by GLM-extended Bayesian phylogeographic inference with heterogeneous evolutionary processes through time. Predictors in the model are shown in Fig. S6. The conditional effect size of each predictor (on a log scale) is presented as mean values with credible intervals. The phylogeographic GLM estimates were obtained from $n = 1845, 1844$, and 1163 viral sequences sampled over 12, 10, and 11 locations for clade 2.3.4.4 (2018–2023), 2.3.4.4 (2010–2017), and 2.3.2.1, respectively. The light blue, darker blue, and red colors indicate clade 2.3.4.4 (2018–2023), 2.3.4.4 (2010–2017), and 2.3.2.1, respectively.

Furthermore, it overlaps well with the bird migration seasonality. Before 2018, most virus lineage movements (14 of 20) show a single temporal peak (Figs. S3.3–3.4, 3A). The peaks of the northward routes overlap with spring bird migration and/or wintering period (upper rows of Fig. 3A, Fig. S3.3). Only one route (Japan-Korea to USA-Canada) overlaps with the summer breeding period. Most southward virus lineage movements peak during the Fall bird migration period, although some peaks continue in November when birds might still be migrating along some routes (lower half of Fig. 3A, Fig. S3.4). Only one route (Europe to Qinghai) overlaps with the wintering period. In summary, in the Northern Hemisphere, virus lineage movements from south to north occur mainly during the wintering period and spring bird migration, while southward virus lineage movements occur mainly during the fall migration period when birds fly to the south. Since 2018, southward virus lineage movements have expanded the time period to late summer and early winter in addition to fall (Fig. S3.6); northward virus lineage movements shift to mainly spring and summer (Fig. S3.5).

This association of seasonality in bird migration and HPAIV H5 lineage movement of clade 2.3.4.4 implies that breeding grounds might be potential pools of HPAIV H5 genetic diversity for southward virus lineage movements associated with bird migration; wintering grounds might play a similar role for the northward viral lineage movements. Additionally, the results show more virus lineage movements during the fall migration (Southward Markov Jump counts: 11.71 ([8, 16] 95% HPDI, before 2018) and 78.76 ([68, 89], 2018–2023) per month, September-November) than the spring migration (Northward Markov Jump counts: 9.83 ([6, 14], before 2018) and 1.15 ([1, 3], 2018–2023) per month, March-May). Virus lineage movements also have higher relative frequency during the fall migration (shown in the higher peak in Fig. 3). Interestingly, birds also migrate in a larger abundance in the fall than during spring, as the population size becomes larger after breeding.

In contrast, for clade 2.3.2.1, virus lineage movements show no seasonal variation across the year (Fig. S3.1–3.2). This difference might be due to the fact that largely wild birds contribute to the virus genetic diversity of clade 2.3.4.4, while that of 2.3.2.1 is not (Fig. S1); this suggests that clade 2.3.4.4 might be endemic in wild birds in some regions, i.e., wild birds might serve as the reservoir.

Which bird orders might be exposed to HPAIV H5 at the origin or destination regions of virus lineage movements? To explore this question, we examined the synchrony of bird migration and virus lineage movements by testing the correlation between the 2 monthly time series. We then used block bootstrapping to test the statistical significance (See Methods). Along some routes, virus lineage movement frequency is correlated with monthly bird order distribution of the origin/destination location (Table S3). We assume that the bird population's birth and mortality are stable, and there is no external intervention. Therefore, the change in bird distribution probability is only due to a positive net flux of incoming individuals through migration or a negative net flux of individuals via migration outside a given area. We highlight the following results for clade 2.3.4.4 between 2018 and July 2023 (Fig. S7, Table S3):

- Southern migration of Ciconiiformes from Europe synchronizes with more frequent viral dispersal from Europe to Africa ($r = -0.47$, 97.5% confidence intervals, CI: [−0.65, −0.29], $p = 3.58 \times 10^{-5}$), suggesting that Ciconniformes might be exposed to clade 2.3.4.4 when they start migrating south from Europe.
- Southern migration of Passeriformes to Africa synchronizes with more frequent viral dispersal from Western Asia to Africa ($r = 0.52$, 97.5% CI: [0.33, 0.79], $p = 5.38 \times 10^{-5}$), suggesting that Passeriformes might be exposed to the viruses when they migrate south to Africa.
- Northern migration of Pelecaniformes from Japan/Korea synchronizes with more frequent viral dispersal from Japan/Korea to Russia ($r = -0.32$, 97.5% CI: [−0.46, −0.03], $p = 8.96 \times 10^{-5}$), suggesting that Pelecaniformes might be exposed to the viruses when they migrate north from Japan/Korea.

For clade 2.3.4.4 before 2018, we detected two virus lineage movement routes, both in Asia, with Anseriformes and Accipitriformes as exposed bird orders, respectively (Fig. 3, Table S3). We do not detect bird orders associated with virus lineage movements of clade 2.3.2.1 (Table S3). Although we cannot conclude any causal relationship from current analyses, the synchrony of virus lineage movement and bird migration suggests the birds might be exposed at the origin/destination location. Future work needs to be conducted to understand the casual relationship.

## Discussion

Here, we report a phylodynamic analysis linking the spatial ecology of avian hosts and HPAIV H5 virus lineage movements. Our results support previous findings on the important role of bird migration in disseminating HPAIV H5 clade 2.3.4.4[21]. We found that the seasonal wild bird migration network is associated with the global diffusion and evolutionary dynamics of HPAIV H5. A previous study found that the 2014/2015 outbreaks of HPAIV H5 in Europe and North America were likely introduced by wild bird migration[21] by comparing the inferred ancestral host-type and location traits of the viral genome sequences[21]. Our study advances this finding by directly integrating the bird migration network into the virus phylogeographic reconstruction. In addition, we found that the inter-regional live poultry trade is not associated with the global HPAIV H5 dispersal, consistent with

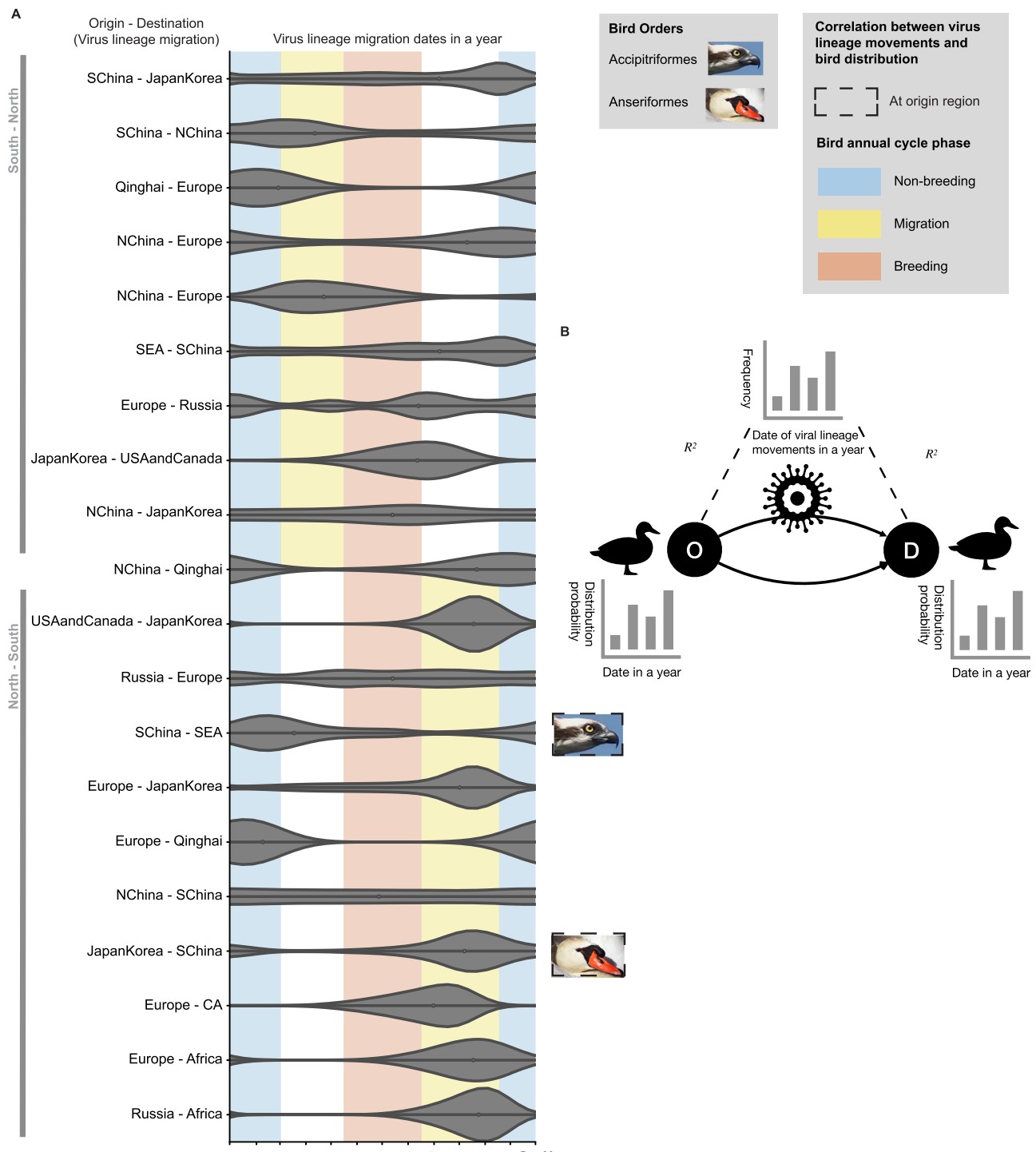

**Fig. 3 | The synchrony of virus lineage movements between regions and bird distribution probability at the regions. A** Probability density distribution of the virus lineage migration throughout the year, between locations summarized from the discrete trait phylogeography of HPAIV H5 clade 2.3.4.4 and the Markov jump counts (Section "Discrete trait phylogeography of HPAIVs and counts of virus lineage migration"). X axis: Virus lineage migration dates in a year; Y axis: origin region - destination region of the virus lineage migration. The width of the violins represents the virus lineage migration probability density. Non-breeding (blue), migration (yellow) and breeding (red) bird annual cycle phases in general are shown in the south-north migration direction and in the north-south migration direction. Boxes around bird photos show the statistically significant correlation of virus lineage movements and bird order distribution at origin, destination or both regions. We used the following images of bird species from the Macaulay Library at the Cornell Lab of Ornithology: *Pandion haliaetus haliaetus* (ML523577271) and *Cygnus olor* (ML72775261). Abbreviations of locations are the same as in Fig. 1. **B** Schematic diagram of cross-correlation analyses of virus lineage movement between two locations (O: origin, D: destination) and the bird distribution probability at each location. Block bootstrapping was used to calculate confidence intervals and two-tailed *p* values. See details in Methods "Discrete trait phylogeography of HPAIVs and counts of virus lineage migration".

previous studies[13,21]. The same previous study found that the international poultry trade's direction is opposite to the global spread direction of HPAIV H5 clade 2.3.4.4[21]. Another study demonstrated large-scale H5N1 transmission dynamics are structured according to different bird flyways and driven by the Anatidae family, while the Phasianidae family, largely representing poultry, is an evolutionary sink[13].

Our results provide insights into relatively under-studied bird orders that may be relevant to HPAIV H5 dispersal. Historically, studies of host-pathogen interaction of AIV have focused on Anseriformes as wild bird hosts. However, many other under-studied orders have been affected by clade 2.3.4.4 recently[33,34]. Interaction of different avian orders might contribute to virus dispersal and local persistence[28]. A previous study showed that host origins of HPAIV H5 reassorted genes include Anseriformes, other groups of wild birds, and domestic poultry[24]. Our results highlight the exposure risk of specific under-studied bird orders in addition to Anseriformes, including Accipitriformes, Ciconiiformes, Passeriformes, and Pelecaniformes. Particularly, Passeriformes are rarely considered in avian influenza studies. It is possible that when Passeriformes have migrated south to Africa, they might be exposed to avian influenza from interacting or sharing habitat with the waterfowls in Africa, as Asia-Eastern Africa is a major migratory flyway where waterbirds, Passeriformes and other species migrate from Asia to Eastern Africa for wintering. However, due to the use of light-level geolocators (GLS) in some Passeriformes tracking studies, our results might be geographically inaccurate given the coarse spatial resolution (~200 km) of GLS location estimates[35].

Our results also highlight the seasonal pattern of virus dispersal of clade 2.3.4.4. Despite the low relative frequency of virus lineage movements during summer breeding and wintering, they may serve as a gene pool for following virus lineage movement during the migration. A previous study emphasizes the important role of the breeding period in inter-species virus transmission in North America[29]. Previous surveillance also shows that LPAI prevalence in waterfowls is higher during the wintering period of Eurasian migratory birds in Africa[36]. Interestingly, clade 2.3.2.1 does not show seasonality in virus lineage movements. The non-seasonal pattern might be because domestic poultry contributes to the virus genetic diversity more than wild birds for clade 2.3.2.1 (Fig. S1), while wild birds contribute much more to the reassortment of the viruses for clade 2.3.4.4 (Fig. S1,[24]). The difference also suggests that clade 2.3.2.1 might not be endemic in wild birds, while wild birds might be the reservoir for clade 2.3.4.4.

By comparing clade 2.3.4.4 before 2018 and after (Fig. S4), our results suggest wild bird migration might be the same mechanism for geographical global dispersal of clade 2.3.4.4 throughout the years; differently, the viruses have become endemic in birds in wider geographical regions since 2018, and breeding time might have become more important in virus dispersal.

Caveats. First, our data in *bird tracking* are much limited in species diversity due to low data availability in the tracking community ([37]). Despite variations in migration behaviors within bird orders, with the data, we are not able to account for the variations. Thus, it should be cautioned that not all species within the bird order we found relevant to the virus lineage movements may be exposed to HPAIV H5. Nevertheless, our tracking data are relevant for avian influenza circulation: the families we use in tracking data (Supplementary Dataset 2) included major wild bird species that have been sampled avian influenza viruses (Table S1). In addition, currently available tracking data may be more abundant in individuals that are easier to capture. Despite the use of the species distribution model to account for the empirical data bias, the uneven abundance of tracking data in species makes the model more accurate for species that are represented by better data. We are also not able to include the migration volume of birds in the migration network (Fig. S6). With the increasing availability of bird tracking data, our framework can further study the association of birds at the species level and virus lineage movements; we may find more

associated species that are relatively under-studied; and we can also advance our binary bird migration network by integrating comprehensive bird movement models[38,39].

Secondly, the *virus genome* data are biased geographically and in host species. The under-sampling of viruses in some areas hugely impacted the inferred phylogeography. For example, we cannot conclude if the inferred viral lineage movement from Europe to Qinghai or Japan-Korea occurs directly or if geographically proximate areas, e.g., central Eurasia, are middle stops of the movement due to under-sampling in central Eurasia. Despite including sampling size in the phylogeographical analysis, we cannot adjust the geographical sampling biases due to the unknown magnitude of infections at locations. Therefore, our results might biased in highlighting the regions with better virus sampling. Fortunately, the sampling efforts in some historically under-sampled and no-sampled areas are growing, e.g., in Australia[40]. In the future, given more extensive and evenly sampled spatial data, we may find historically under-studied regions have risks of virus lineage movements. To avoid the effect of geographical sampling bias in phylogeography (although not for no sampling areas), our framework can also easily adapt structured coalescent methods, e.g., MASCOT[41], when being applied to less geographical locations or less sequence data. In addition to the geographical sampling bias in virus genome data, our results may be affected by poultry-sampled viruses. To preserve genetic diversity, we included virus samples from domestic poultry when inferring virus diffusion. Therefore, some patterns in the results could reflect virus transmission between domestic poultry and spillover from wild birds to poultry rather than bird migratory patterns. In the future, with better-recorded host species in virus genome data, we can better understand the interaction and contribution of poultry and wild birds in virus evolution and dispersal, including hosts' contribution to reassortment. Currently, we only considered the HA gene when inferring AIV diffusion and evolution. HA is a key gene in influenza viruses, as it is the receptor-binding and membrane fusion glycoprotein of influenza virus and the target for infectivity-neutralizing antibodies[42]. However, the reassortment events of all internal genes are also important in the dispersal and evolution of HPAIV H5[24].

Finally, when analyzing the statistical association between bird distribution and virus lineage movements, we do not include *time lags* between the two time series. Therefore, some bird orders that synchronize with virus lineage movements with time lags may not be detected in our current analyses. Those birds may be exposed to the viruses because of inter-species transmission or environmental transmission if they are possible transmission routes among the birds - it should be noted, however, that currently, little is known about the mechanisms of transmission in wild birds, particularly inter-species transmission among seabirds. With finer-scale bird tracking data on movements other than migratory movements[43], future studies could better integrate avian biology in understanding the inter-species transmission mechanism. With longitudinal bird tracking data at locations and virus sampling from the birds, our analysis framework could be advanced to understand the time lags of the synchrony and different roles of bird species in HPAIV H5 dispersal (Table S4). Another limitation in our methods is that bird migration networks in our analyses are binary. Therefore, when we model a non-bird-migration time period in Fig. 2, the binary network does not represent no migration; instead, it represents that all the relative migration rates between locations are the same.

In conclusion, allocating more resources for global surveillance of avian influenza viruses in wild birds would enhance our ability to tackle the challenges of more virulent and transmissible HPAIV H5 spreading in wild birds. To achieve this goal, it is critical to understand "where and in which bird species surveillance is most needed and could have the greatest impact"[14]. Given sufficient data in the future, our framework could help conservation and public health policy-making in

designing monitoring and surveillance strategies. More collaboration is needed between ornithologists, movement ecologists, bird conservation experts, avian influenza epidemiologists, disease ecologists and virologists on many aspects, including collaborative data collection/surveillance of AIV and data sharing. For example, if studies were to simultaneously obtain the movement tracking of bird populations and their serology and virology surveillance data, then they could link the bird movement directly with the virus transmission and dispersal. In addition, we need more AIV samples from water bodies to better understand environmental transmission. With such data, we would be able to understand the viral transmission at local scales and therefore develop disease models for bird conservation and potential zoonotic threats.

## Methods

### Viral sequence data and time-scaled phylogeny of HPAIVs
To infer the phylogeny of avian influenza HPAIV H5 viruses, we accessed sequences of HA genes, NA genes and six internal gene segments from GISAID (Global Initiative on Sharing All Influenza Data[44–46]). Using the sequences, we estimated a maximum likelihood phylogeny (Fig. S1) for each gene segment, respectively, under a GTR+$\Gamma$ nucleotide substitution model, with the randomly selected strains as representatives, by FastTree v2.1.4[47]. Genotypes of internal gene segments (Fig. S1) were defined by clustering patterns with background sequences in a previous study[48]. On the phylogeny, the viruses with internal genes from wild birds, e.g., clade 2.3.2.1 and clade 2.3.4.4, showed wider geographical spread[21,49], compared to poultry viruses, e.g., clade 2.3.4.1 and clade 2.2, despite the high similarity of their HA genes. This demonstrates the importance of gene reassortment in the evolution and transmission of HPAIVs.

In this study, we focus on clade 2.3.4.4 and clade 2.3.2.1. We inferred their time-scaled phylogenies of HA genes. Before the inference, to test for the presence of phylogenetic temporal structure, we generated a scatterplot of root-to-tip genetic divergence against sampling date by TempEst v1.5[50]. Strong phylogenetic temporal structure was detected in the phylogeny of each clade (Fig. S5). The final datasets were i) 1163 HA sequences of clade 2.3.2.1; ii) 1844 HA sequences of clade 2.3.4.4 (2010–2017); iii) 1845 HA sequences of clade 2.3.4.4 (2018–2023, after down-sampling). For clade 2.3.4.4 (2018–2023), we randomly down-sampled over-represented locations (with sequences more than 250) to 262 for each location to get a similar size of sequences across locations and to get a similar size of total sequences as previous clade 2.3.4.4. The spatial and temporal distribution of the sequences is shown in Fig. S2.

Time-resolved HA phylogenies were estimated using the Markov chain Monte Carlo (MCMC) approach implemented in BEAST v1.10.4[51] with the BEAGLE library[52]. We used an uncorrelated lognormal relaxed molecular clock model[53], the SRD06 nucleotide substitution model[54] and the Gaussian Markov random field Bayesian Skyride coalescent tree prior[55]. For each dataset, MCMC chains were run for 300 million (clade 2.3.2.1) and 400 million (clade 2.3.4.4) generations with burn-in of 10%, sampling every 10,000 steps. Convergence of MCMC chains was checked with Tracer v1.7[56]. A set of 1000 trees for each clade was subsampled from the MCMC chain and used as an empirical tree distribution for the subsequent analysis.

### Animal mobility networks and their contribution to HPAIV phylogeography
The bird migration network (Fig. S6) was summarized by searching publicly available migration data on Movebank. An edge between two locations in the network exists if any migration tracking record shows bird migration originating from the start location to the end location. The location-wise live poultry trade values were summed up from country-wise import and export of the live poultry recorded on United Nations Comtrade Database (comtrade.un.org/data/). We accessed the total net weight and trade value from 1996 to 2016 of live poultry, including fowls of the species Gallus domesticus, ducks, geese, turkeys and guinea fowls. Since there are no accessible data of the within-country poultry trade in China, we adapted the inferred poultry trade accessibility between provinces of China from a previous study[57]. Based on the ratio of the inferred accessibility and the empirical trade value between Hong Kong SAR and mainland China, we scaled all the accessibility to the trade value flows among Qinghai, North China and South China.

With the summarized seasonal-varying bird migration network, we statistically quantified the contribution of wild bird migration to avian influenza virus dispersal. We applied the GLM extended Bayesian phylogeography inference[31] with the 1000 empirical trees as the input. The aggregated locations in the previous discrete trait phylogeography were still used. The epoch model[32] was used to account for the time heterogeneity of the contribution. For each clade and each predictor group, MCMC chains were run for 100 million generations with burn-in of 10%, sampling every 10,000 steps. Similarly, we assessed the convergence of the chains in Tracer v1.7[56].

### Wild bird movement tracking and distribution modeling
To assemble the global wild bird observation data, we accessed the worldwide bird tracking data from Movebank in 2021. This dataset amassed from 53 studies across the world[58–131]. The Movebank study ID, name, principal investigator, and contact person are listed in Supplementary Dataset 2. The dataset is collected by various research groups, and by various sensors, including Global Position System (GPS), Argos, bird ring, radio transmitter, solar geo-locator, and natural mark. It covers over 3542 individual birds (class: Ave), including ten orders and 95 species (Supplementary Dataset 2). For further modeling the migration of the wild birds belonging to different orders, we excluded the observation data on Movebank of Cuculiformes, Caprimulgiformes, Strigiformes, Columbiformes, Phoenicopteriformes, Piciformes, Sphenisciformes, and Procellariiformes, given their paucity and geographically restricted distribution. Additionally, we accessed GPS tracking data of 193 individuals, including 5 orders and 12 Species between 2006 and 2019 in China from a previous study (ref. 132). Accordingly, we combined the data from China with those on Movebank (Supplementary Dataset 2) and finalized a bird observation dataset consisting of 10 orders and 96 species.

To model the wild bird distribution throughout a year, we developed a model framework based on the species distribution model (SDM). The response variable of the model is bird occurrence (1: presence; 0: pseudo-absence). The independent variables are 20 well-studied environmental predictors[133–137] (Table S1). We divide the globe into 1-km resolution geographical cells for each month. For each cell, the value of the dependent variable is 1 if there is any observation of an individual in the target order in that month in the bird tracking data, otherwise 0. Furthermore, to infer the probability of bird occurrence between 0 and 1 for each cell, we trained a XGBoost binary classification model[138] for each bird order, respectively. The method is adapted from a previous bird migration model[139]. We used true-presence and pseudo-absence data (marked as 1 and 0 respectively). We fitted the distribution of birds which manifest as true-presence data and pseudo-absence data. We randomly divided 67% of the data as the training set and the other 33% as the test set. The model finally outputs the probability of the distribution of migratory birds in each month across years. The accuracy was evaluated by the area under the curve in a test set of the ten orders: Pelecaniformes (0.97), Gruiformes (0.97), Passeriformes (0.97), Suliformes (0.98), Ciconiiformes (0.92), Falconiformes (0.98), Charadriiformes (0.94), Anseriformes (0.90), Accipitriformes (0.90). The modeled wild bird distribution was applied in the subsequent analysis to identify key bird orders associated with the global viral dispersal (see Methods) and local virus emergence.

## Discrete trait phylogeography of HPAIVs and counts of virus lineage migration

Based on empirical phylogenies, we used a non-reversible discrete-state continuous-time Markov chain model and a Bayesian stochastic search variable selection (BSSVS) approach[30] to infer the viral diffusion among locations: i) the most probable locations of the ancestral nodes in the phylogeny and ii) the history and rates of lineage movement among locations[30]. Sampled countries were divided into 10 locations: Africa, Central Asia, Europe, Japan-Korea, North China, South China, Qinghai, Russia, Southeast Asia and USA-Canada. This regional categorization was done according to the major wild bird breeding areas. Furthermore, to estimate the viral gene flows between locations, we used a robust counting approach[140,141] to count virus lineage migration events. The basic idea is to count the expected number of lineage movements (Markov jumps) between the locations along the phylogeny branches, as applied in previous studies[142–146]. The count of virus lineage migration events was used for further analysis below. We also summarized monthly frequency distribution of the virus lineage migration for each pathway (Fig. S3).

To target the key bird orders for each location, we explored the association of wild bird distribution across a year and the virus diffusion. The monthly wild bird distribution probability at each location is generated based on the location's geographical coordinates on the modeled bird distribution probability raster map. We then applied block bootstrapping to generate samples of monthly virus lineage movement counts from all events while keeping the year and the month of the event. The number of samples is decided by $\frac{20}{0.05} \times$ the number of routes $\times$ the number of bird orders. We summarized the counts by month in each sample to get a monthly count sample. We calculated Pearson's correlation ($r$) for each sample between the virus lineage migration of each route and each bird order distribution at the origin/destination region, respectively. We define the two-tailed $p$ value by doubling the proportion of the bootstrap samples with either $r > 0$ or $r < 0$ (whichever is smaller). To account for multiple comparisons of 9 bird orders, $n$ routes, and 2 locations (origin/destination) on each route, we use $p$ value $< \frac{0.05}{9 \times 2 \times n}$ to define the statistical significance in the correlations. When bird distribution at the origin/destination is correlated with the virus lineage movements positively or negatively, we consider the bird order distribution at the origin/destination correlated with the virus lineage movements.

### Reporting summary

Further information on research design is available in the Nature Portfolio Reporting Summary linked to this article.

## Data availability

We provide Movebank Study ID (unique searchable identifier) and relevant metadata information for bird tracking data available on Movebank database (movebank.org) in Supplementary Dataset 2. We provide accession ID and relevant metadata for virus genomic data available on GISAID database (gisaid.org) in Supplementary Dataset 1. Live poultry trade data are available on the United Nations Comtrade Database (comtrade.un.org/data/). Live poultry trade data within China and the country-level poultry population size data used in phylogeographic analyses are available on the GitHub Repository[147]. The environmental variable data used in the species distribution model are publicly available on WorldClim (www.worldclim.com/version2), NASA ARC ECOCAST GIMMS NDVI3g v1p0: Version 1.0. (iridl.ldeo.columbia.edu/SOURCES/.NASA/.ARC/.ECOCAST/.GIMMS/.NDVI3g/.v1p0/index.html?Set-Language=en), Global 1-km Consensus Land Cover (www.earthenv.org/landcover) and LP DAVV and Data Products (lpdaac.usgs.gov/about/citing_lp_daac_and_data).

## Code availability

All code and scripts are provided on GitHub Repository[147].

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

## Acknowledgements

We gratefully acknowledge all data contributors of virus genome sequences, i.e., the authors and their originating laboratories responsible for obtaining the specimens, and their submitting laboratories for generating the genetic sequence and metadata and sharing via the GISAID Initiative, on which this research is based.We also gratefully acknowledge all bird tracking data contributors, i.e., the authors and other researchers in their originating groups collecting the data and metadata and sharing via Movebank, on which this research is based. The principle investigator, contact person, citation, or data repository DOI are listed in Supplementary Dataset 2.We thank Qizhong Wu and Joint Center for Earth System Modeling and High Performance Computing, Beijing Normal University, for providing computing resources for time-scaled phylogenetic analyses. We thank Jing Yu for assisting in assembling international poultry trade data. We thank Bram Vrancken for assisting with phylogenetic analyses. We thank insightful comments from Andrew Rambaut and the Virus Club Group at the University of Edinburgh, Grenfell lab at Princeton University, the Influenza group at Francis Crick Institute, Olga Alexandrou at Society for the Protection of Prespa, Yonghong Liu at Chinese Center for Disease Control and Prevention, Yidan Li (formerly at Beijing Normal University), and Nils Stenseth at the University of Oslo.Q.Y. acknowledges funding from the Center for Health and Wellbeing, Princeton University, via Graduate Funding for Health-Focused Research. P.L. acknowledges support from the European Union's Horizon 2020 research and innovation program (grant agreement no. 725422-ReservoirDOCS), from the Wellcome Trust through project 206298/Z/17/Z and from the European Union's Horizon 2020 project MOOD (grant agreement no. 874850). S.W.P. was supported by a Charlotte Elizabeth Procter Fellowship of Princeton University. J.L.H.T. is supported by a Yeotown Scholarship from New College, University of Oxford. Y.H. is supported by a Masason Foundation Fellowship, a Honjo International Fellowship, and a Centennial Fellowship. O.G.P. was supported by the UKRI GCRF One Health Poultry Hub (grant no. B/S011269/1). G.Z. is supported by National Natural Science Foundation of China (32070530). H.T. is supported by National Key Research and Development Program of China (2022YFC2303803), National Natural Science Foundation of China (82073616), Fundamental Research Funds for the Central Universities (2233300001), and BNU-FGS Global Environmental Change Program (No.2023-GC-ZYTS-11). B.T.G. is supported by Princeton Catalyst Initiative, Princeton University, and High Meadows Environmental Institute, Princeton University. The opinions expressed in this article are those of the authors and do not reflect the view of the National Institutes of Health, the Department of Health and Human Services, or the United States government.

## Author contributions

Q.Y., B.T.G., H.T., O.G.P., and G.Z. contributed to the design of the work. Q.Y., G.Z., H.T., L.D., Y.C., and B.L. contributed to the data acquisition. Q.Y., B.W., and P.L. contributed to the data analysis. Q.Y., P.L., L.D., T.M., R.A.W., F.G., S.W.P., N.L., J.T., S.B., Y.H., G.Z., O.G.P., H.T., and B.T.G. contributed to the data interpretation. Q.Y. and B.T.G. drafted the initial manuscript draft. All authors contributed to the critical revision of the manuscript and approved the final version of the manuscript.

## Competing interests

The authors declare no competing interests.

## Additional information

¹Department of Ecology and Evolutionary Biology, Princeton University, Princeton, NJ, USA. ²State Key Laboratory of Remote Sensing Science, Center for Global Change and Public Health, Faculty of Geographical Science, Beijing Normal University, Beijing, China. ³Department of Microbiology, Immunology and Transplantation, Rega Institute, KU Leuven, Leuven, Belgium. ⁴College of Life Sciences, Beijing Normal University, Beijing, China. ⁵Princeton School of Public

and International Affairs, Princeton University, Princeton, NJ, USA. [6]Fogarty International Center, U.S. National Institutes of Health, Bethesda, MD, USA. [7]Animal and Plant Health Agency-Weybridge, OIE/FAO International Reference Laboratory for Avian Influenza, Swine Influenza and Newcastle Disease Virus, Department of Virology, Addlestone, UK. [8]Department of Pathobiology and Population Science, Royal Veterinary College, London, UK. [9]Department of Biology, University of Oxford, Oxford, UK. [10]State Key Laboratory of Biocontrol, School of Ecology, Shenzhen Campus of Sun Yat-sen University, Shenzhen, Guangdong, China. [11]Institute for Disease Modeling, Bill and Melinda Gates Foundation, Seattle, WA, USA. [12]Key Laboratory of Forest Protection of National Forestry and Grassland Administration, Ecology and Nature Conservation Institute, Chinese Academy of Forestry, National Bird Banding Center of China, Beijing, China. [13]These authors jointly supervised this work: Guogang Zhang, Oliver G. Pybus, Huaiyu Tian, Bryan Grenfell. ✉e-mail: qiqiy@princeton.edu; tianhuaiyu@gmail.com; grenfell@princeton.edu

