## [Peer Review File · Nature Communications]

Synchrony of bird migration with global dispersal of avian influenza reveals exposed bird ordersREVIEWER COMMENTS

Reviewer #1 (Remarks to the Author):

In their paper, the authors have attempted to explain the role that wild birds play in the global dispersal of highly pathogenic avian influenza (HPAI) viruses of the Gs/GD/96 lineage. By integrating bird migration ecology in phylodynamic analyses, the authors have demonstrated that the seasonal wild bird migration network is associated with the global diffusion and evolutionary dynamics of HPAI H5. Although other studies have already clarified the crucial role of migratory birds in the transcontinental dissemination dynamics of the H5 virus, the multidisciplinary approach described in this study provides a higher level of resolution in the reconstruction of viral dissemination dynamics and provides new insights into the bird orders potentially involved in viral gene flows. Overall, the paper is well written but a few changes are recommended:

- The analyses are based on a dataset that includes H5Nx sequences up to year 2018 but does not comprise the most recent ones. The global situation for avian influenza has seriously changed in recent years. The virus has become endemic in wild birds populations resident in different geographical areas. This has completely disrupted the epidemiological dynamics of the virus with the occurrence of epidemic events in wild birds at any time of the year, including the summer months. It would be important for the authors to discuss the possible implications of such changes in viral dynamics and the consequent limitations of the study.

- As the authors reported in the Material and Methods section, most analyses were performed also using sequences of clade 2.3.2.1. The Introduction, Results and Discussion focus on clade 2.3.4.4, which may be likely attributable to reasons of epidemiological relevance. However, as this is a retrospective study, it would be beneficial to clarify whether any differences in the dynamics between the two clades were observed. Even if not globally spread as clade 2.3.4.4, we have to keep in mind that clade 2.3.2.1 is still circulating in many Asian territories and has recently caused a spill-over in humans in Cambodia.

-Paragraph 2.2. Passeriformes have been included among the "vulnerable" migratory bird orders. As this is an unexpected data for AIV, I believe it is worth being discussed further.

Minor revisions:

- The title of the paper refers to "implications" for vulnerable bird orders, although this aspect is not explored in the manuscript and may mislead the reader. I suggest either to eliminate this term from the title or to expand the Discussion section.

- In the Introduction, the authors refer to the "re-emergence" of HPAIVs; in truth, I am not sure it is correct to talk about re-emergence considering that the virus has been circulating and evolving since 2008. It would be better to talk about resurgence or write the sentence as follows "The highly pathogenic avian influenza viruses (HPAIVs) subtype H5 clade 2.3.4.4 have caused unprecedented large numbers of wild bird deaths worldwide since 2014"

Reviewer #2 (Remarks to the Author):

This is an interesting paper on an important current topic, seeking to show how bird migration has explained the spread and incursions of the recent and ongoing highly pathogenic H5 2.3.4.4 outbreaks in different countries. The authors use data on bird migration from public sources (movebank), poultry trade data, and viral sequence data from GISAID in a global analyses covering at least 10 years. The paper is generally well written and the results look justified given the methods, so I have a few comments.

Minor and technical comments

1. You seem to use clade 2.3.2.1 as a comparator to 2.3.4.4 but don't really talk about this very much in the main text. Perhaps include a little in the introduction and discussion on this (a

sentence or so?), e.g. is 2.3.2.1 is more 'poultry' driven than 2.3.4.4 ? (Figure 1B suggests not).

2. The sequence data is used to make time-scaled phylogenies, and discrete location and bird species traits are mapped onto the trees. I think you then use a binary network (line 257) in a discrete geographic region phylogenetic GLM framework but also include origin and destination sample size (non-binary) and shown in Figure 1B ? But did you (and why did you) use only one binary wild bird migration network (all birds species together) vs making a network for different species types ? I ask because the sea-birds / shore-birds might have a very different network than anseriformes (e.g. sea-birds / shore-birds go along the coasts and seas, not large distances inland). And the binary network just connects the few discrete geographic broad regions, but what is the correlation between the bird migration network and distance between the regions, and so did you also try to include distance as a predictor ? Also, the bird data is composed into presence and pseudo-absence on a grid (1km²), per species, is this used and aggregated to make the binary network ? this seems a lot of work to get a binary network between large regions, but I notice that you use this data in a temporal phylodynamic analyses.

3. Line 136 - if the sequences are highly clustered by region (large region), then why not use BEAST2/MASCOT or other structured coalescent framework, as this might help mitigate the biased rates results problems which are reportedly more evident in a discrete traits analysis ? Note I'm not suggesting that you actually do this, because it might be that your datasets are too large at well over 1000 sequences each to get this to work in a sufficiently timely manner and that the results would not be sufficiently different anyway.

4. Line 160 - did you try downsampling but maintaining diversity as much as possible ? (e.g. X per bird species-type per month per country). Looks to me like clade 2.3.4.4 results in Figure 1B are affected by relatively over-sampled poultry ? Or is this just because the poultry sampling is also reflective of the the poultry population size ? (or number of reported outbreaks because HPAI is notifiable).

5. Lines 198, 208 - 216 - are these results affected by using only one binary bird movement network ? (See point 2).

Also on line 202 - how many events are there from Europe -> Qinghai in your dataset ? Are you speculating on the basis of essentially one event [branch] with under sampled / unobserved in between transmissions, how long is that branch (in time) ? [I note in the text you put 'we cannot conclude that'. which I agree with].

6. Line 382 Spelling - should be Markov Jumps (no e).

7. Figure S5 - Strong temporal signal in Tempest of HA

Are you sure you input a non-timescaled tree (e.g. maximum likelihood tree) into Tempest to generate these plots ? Because it looks to me like you have included the plots using the BEAST MCC tree; since the root-to-tip distance (y-axis) is on the same time scale as the x-axis, which is not correct; you should use a non-time scaled tree for these plots.

Reviewer #3 (Remarks to the Author):

Highly pathogenic avian influenza virus H5 clade 2.3.4.4 has caused worldwide outbreaks in domestic poultry and increasing deaths of diverse species of wild birds since 2014, with massive outbreaks in novel wild bird species since 2021. In this context, the current study is proposing to address how seasonal wild bird migration facilitates global virus dispersal, and which avian species are exposed to HPAI H5 clade 2.3.4.4 viruses, with the idea of potentially shedding light on ongoing global outbreaks, with high relevance for conservation and global epidemiology. This is done by conducting phylodynamic analyses based on empirical data of bird movement tracking and virus genome sequences. Although the topic is of great interest, a large amount of data was processed and the results appear to make sense, the study is disappointing because (1) it does not put the results in context of the recent large scale outbreaks (for which much data is already

available, notably in terms of wild bird species affected, but also of the virus clades involved, e.g., EFSA 2022), (2) the presentation of some of the results make them difficult to understand (the whole part on the 'correlation between the virus lineage migration and the bird probability distribution at origin and destination regions' is confusing), and most importantly it suffers weaknesses due to the data used, notably on the avian ecology side of things. I concur with the authors that it is of importance to integrate wild bird ecology in avian influenza studies, but I think this is not done convincingly enough in the current manuscript. The 'Caveat' part (page 9) is welcome and a critical part of the manuscript, but the call for more data would gain to include some suggestions about sampling design issues. Also, the manuscript would gain in discussing more clearly how the methods used may be affected by sampling limitation and how this might change with increasing availability of different types of data.

In the introduction (lines 83-86), it is stated 'Integrating host movement in studying HPAIV dispersal is important while challenging. One challenge is insufficient bird movement data, which causes that previous global-scale studies [23] cannot account for the high variation in bird behaviours across species and locations'. But does the current study do a good job at soundly considering variability in bird movement across species and locations? Using what can look as an impressive amount of data may not be sufficient to make reliable inference. The movement data used in the analyses is mostly the data freely available from Movebank, a repository where researchers declare and manage their animal movement data, some of them making their data available. Only a small fraction of the gathered data worldwide are freely accessible that way, and the data that was used, from looking at the content of the data set (and as described in the text), appears to be limited to only a very small and biased sample of the actual data acquired worldwide on bird movements. For instance, data from a total of 53 studies of all bird species was considered from Movebank in the current study (lines 308-309), leading to the use of movement data for a total of less than 4000 individuals, while in a recent paper reviewing tracking data only in seabirds, Bernard et al. (2021) say 'To optimize future tracking efforts, we performed a global assessment of seabird tracking data. We identified and mined 689 seabird tracking studies, reporting on > 28,000 individuals of 216 species from 17 families over the last four decades.' And of course seabirds only represent a small part of wild birds. A critical issue is thus what proportion of species were used in the current study compared to wild bird species or bird movement studies relevant for avian influenza circulation, and was the tracking data used in the analyses meaningful/biased? Bird (and animals/humans) do different types of movements and each of those may potentially have very different meaning for the risk of disease spread (e.g., Boulinier 2023). Many bird species are for instance central place foragers during breeding and this is when many are temporally equipped with GPS loggers to track those foraging movements; conversely, GLS loggers have been more used to explore large-scale migratory movements, which may be critical for some epidemiology questions, but they have much lower spatial precision). The data set used to characterize habitat use by bird orders is thus relatively small while very heterogeneous in terms of the types of considered movements (within a breeding season, across years...), which likely limits what can be interpreted from the results.

Regarding the first part on how seasonal bird migration associates with global HPAIV H5 dispersal, it is worrying that sample size is retained in the model as a key factor. As the authors stress, this could mean the patterns were affected by sampling bias, but it is unclear how the inclusion of sample size as a factor was sufficient to address potential biases, especially if there were (likely) interactions between sample sizes and spatial locations of more or less critical relevance for virus transmission. Also, when looking at the whole data set used on virus subtypes, it appears for instance that most data come from Anatids, while e.g. only 4 virus samples for clade 2.3.4.4 (over a total of 1845) come individuals of the order Sulliformes: how does this relate to key reported results about Sulliformes? And the same for other orders?

Overall, more information should be provided on the potential sampling biases and their implications: what proportion of species of which bird orders were considered? What were the feeding habits of the considered species within each orders? Raptors have common habits of feeding on other animals, notably warm-blooded ones, but within other bird orders the behaviours may be more heterogeneous, e.g., in some orders most species forage on food unlikely to be associated to exposure to the viruses, while others feeding on bird carrions and preys. This may affect within species transmission as well as among species transmission. Given the relatively small numbers of bird studies considered, this is an issue that needs to be addressed.

Line 144: Live poultry trade was not included in terms of (net) fluxes of birds? How could that affect the results?

Line 239: strange wording: 'Historically, Anseriformes have been the focus of wild bird hosts when studying host-pathogen interaction in AIV studies'.

Lines 299-301: yes, indeed.

Line 394: how is it accounted for that many lag times were considered?

In the reference list, some references are repeated (63, 64 ?)

It is strange that references to movement studies on Antarctic petrels are listed (48, 61, 104) while it is said in the Methods that studies on Procellariiformes were excluded (lines 314-318)(the relevance of Antarctic habitat is also unclear for the current study). Whether the data was eventually used or not (I understand not), it illustrates the strong heterogeneity of the considered data set on bird movements.

Cited references

Boulinier, T. 2023. Avian influenza spread and seabird movements between colonies. *Trends in Ecology & Evolution* 38: 391-395.

Bernard, A., Rodrigues, A.S.L., Cazalis, V. & Grémillet, D. 2021. Toward a global strategy for seabird tracking. *Conservation Biology* 14: e12804.

General response:

We appreciate all the comments and suggestions from the reviewers for improving the manuscript. We have taken all the comments into consideration and revised our manuscript accordingly. Specifically, to address the comment on including recent data of clade 2.3.4.4, we have conducted complementary analyses of clade 2.3.4.4 using the most recent virus genome data from 2018 to July 2023. Based on the results, we have discussed the implications of the findings in light of more recent outbreaks of avian influenza. **Essentially, the key finding of the importance of bird migration is unaffected (Figure 2).** We have also further discussed the limitations of the analysis given the data used, especially regarding the bird tracking data and the sampling bias in virus genome data.

We have paid particular attention to statistical concerns raised by Reviewer 3. We have conducted additional statistical analyses to further substantiate our findings: 1) controlling for sample size in the generalized linear model (GLM) extension of Bayesian phylogeographic analysis (Table S4) and 2) bootstrapping to calculate correlations of bird migration and virus lineage movement (Figure 3, Table S7).

In addition, we have updated the bird migration network (Figure S6) to include South America (shown as LA on Figure S6). When going through the original tracking data, we have also added a few routes that were missing: Russia <-> South China, Canada/USA <-> Japan/Korea, Central Asia -> Mongolia. We also have corrected the binary matrices of bird migration as un-log-transformed predictors in the GLM extension of Bayesian phylogeographic analysis. The updated results show that bird migration is the major contributing predictor for the virus lineage movements of clade 2.3.4.4, while it is not for clade 2.3.2.1. This difference between clades is consistent with the difference in the seasonal pattern of virus lineage movement of the two clades (Figure S3.1-3.6).

Finally, we believe the previous term ‘vulnerable bird order’ caused some common confusion for reviewers – we initially defined the ‘vulnerable’ bird orders as bird orders that are more likely to be exposed to the HPAIV. We have replaced ‘vulnerable’ to ‘exposed’ across the manuscript including the title.

Please see our point-by-point responses to the reviewer’s comments below.

Reviewer #1 (Remarks to the Author):

In their paper, the authors have attempted to explain the role that wild birds play in the global dispersal of highly pathogenic avian influenza (HPAI) viruses of the Gs/GD/96 lineage. By integrating bird migration ecology in phylodynamic analyses, the authors have demonstrated that the seasonal wild bird migration network is associated with the global diffusion and evolutionary dynamics of HPAI H5. Although other studies have already clarified the crucial role of migratory birds in the transcontinental dissemination dynamics of the H5 virus, the multidisciplinary approach described in this study provides a higher level of resolution in the reconstruction of viral dissemination dynamics and provides new insights into the bird orders potentially involved in viral gene flows. Overall, the paper is well written but a few changes are recommended:

- The analyses are based on a dataset that includes H5Nx sequences up to year 2018 but does not comprise the most recent ones. The global situation for avian influenza has seriously changed in recent years. The virus has become endemic in wild birds populations resident in different geographical areas. This has completely disrupted the epidemiological dynamics of the virus with the occurrence of epidemic events in wild birds at any time of the year, including the summer months. It would be important for the authors to discuss the possible implications of such changes in viral dynamics and the consequent limitations of the study.

[Response] We agree on the importance of including the recent H5Nx sequences given the seriously changed global epidemiology of avian influenza. Therefore, we've analyzed the recent H5Nx sequences (year 2018 to July 2023) using same methods (Line 371-375), and have updated our results (Page 4-6) and discussion (Line 278-282) accordingly.

The recent virus samples include three new regions: South America, Mongolia, and Western Asia, compared to samples before 2018, reflecting the wider spread of the virus. The phylogeography shows that most virus diversity is maintained in Russia, Africa and Europe as demonstrated by the location estimates associated with the trunk of the phylogeography (Figure 1B), while China exhibited the largest virus diversity before 2018 (Figure 1A). This reflects that the viruses have become endemic in different geographical areas.

Importantly, the virus lineage movements show strong seasonality (Figure S3.5-3.6). As in our previous analyses, the wild bird migration network also remains the major contributing factor (log conditional effect size mean: 4.71, HPDI: 3.60 - 6.22, posterior probability > 0.999 and Bayes factor >16565).

In terms of viral dynamics, we observe different patterns from previous years: northward virus lineage movements peak during spring and summer; southward movements peak during late summer, fall and early winter. This reflects the occurrence of epidemic events in wild birds at different times across the year, as the reviewer pointed out.

- As the authors reported in the Material and Methods section, most analyses were performed also using sequences of clade 2.3.2.1. The Introduction, Results and Discussion focus on clade 2.3.4.4, which may be likely attributable to reasons of epidemiological relevance. However, as

this is a retrospective study, it would be beneficial to clarify whether any differences in the dynamics between the two clades were observed. Even if not globally spread as clade 2.3.4.4, we have to keep in mind that clade 2.3.2.1 is still circulating in many Asian territories and has recently caused a spill-over in humans in Cambodia.

[Response] We agree with the reviewer on the importance of including clade 2.3.2.1 in our study. We have clarified differences in the dynamics between clade 2.3.2.1 and clade 2.3.4.4 in Results accordingly. We have also included discussion (Line 266-277) and introduction (Line 57-61) about clade 2.3.2.1.

For 2.3.4.4, the wild bird migration network is the major factor contributing to the virus lineage movements (Figure 2); it is not for clade 2.3.2.1. Clade 2.3.2.1 does not show seasonality in virus lineage movements (Figure S3.1-3.2), while clade 2.3.4.4 both before and after 2018 does (Figure S3.3-3.6). This difference suggests that wild birds largely contribute to the virus genetic diversity in clade 2.3.4.4, while this is not the case for 2.3.2.1; and that clade 2.3.4.4 might be endemic in wild birds in some regions, i.e., wild birds might serve as the reservoir.

This hypothesis is supported by some evidence. First, as clade 2.3.2.1 has been endemic in domestic poultry in many regions, it is possible that both migratory birds and local poultry contribute to the virus genetic diversity. In addition, our phylogenetic analysis of the internal genes shows that wild birds may contribute much more to the reassortment of the viruses for clade 2.3.4.4 compared to clade 2.3.2.1 (Figure S1). The importance of internal genes in demonstrating the contribution of wild birds to the evolution of clade 2.3.4.4 has also been previously shown (Lycett et al. 2020).

Despite the limitation of our phylogeographic analysis in its consideration of recombination or reassortment in virus genetic diversity (as we discussed on Line 321-325), it reflects the combined virus genetic diversity (via mutation and reassortment/recombination). Future work is needed to distinguish the relative contribution of different evolutionary mechanisms by different hosts.

In summary, wild bird migration mainly contributes to the inter-regional dispersal of clade 2.3.2.1 similar to clade 2.3.4.4; However, domestic poultry might contribute to the virus genetic diversity more than wild birds for clade 2.3.2.1 and it is not endemic in wild birds, while wild birds might be the reservoir for clade 2.3.4.4.

Lycett, S. J., Pohlmann, A., Staubach, C., Caliendo, V., Woolhouse, M., Beer, M., Kuiken, T., & Palese, P. (2020). Genesis and spread of multiple reassortants during the 2016/2017 H5 avian influenza epidemic in Eurasia. *Proceedings of the National Academy of Sciences of the United States of America*, 117(34), 20814–20825.

-Paragraph 2.2. Passeriformes have been included among the “vulnerable” migratory bird orders. As this is an unexpected data for AIV, I believe it is worth being discussed further.

[Response] We have revised our analysis based on other reviewers’ suggestions and the updated results show immigration of Passeriformes to Africa is associated with virus lineage movement

in clade 2.3.4.4 (2018-2023), specifically from Western Asia to Africa. It is indeed a surprising finding as the reviewer suggests, because avian influenza is only infrequently reported in Passeriformes. However, it is possible that Asia-Eastern Africa represents major migratory flyway for waterbirds, Passeriformes and other species to migrate from Asia to Eastern Africa to winter. They might get infected by interacting or sharing habitat with the waterfowls in Africa. However, inter-species interaction is beyond the scope of our analyses as we discussed in the manuscript (Line 326-336). However, it should be cautioned that, due to the common use of light-level geolocators (GLS) in Passeriformes tracking studies, our results might be geographically inaccurate given the coarse spatial resolution (~200 km) of GLS location estimates.

Historically, Passeriformes are under-studied in both avian influenza studies and bird tracking studies. In the future, longitudinal surveillance of avian influenza and tracking of Passeriformes, e.g., at wintering sites of the Asia-Eastern Africa flyway, will help to understand the risk of exposure and the role of Passeriformes in avian influenza spread.

We have included related discussion in manuscript (Line 258-265).

Minor revisions:

-The title of the paper refers to “implications” for vulnerable bird orders, although this aspect is not explored in the manuscript and may mislead the reader. I suggest either to eliminate this term from the title or to expand the Discussion section.

[Response] We have removed this term from the title. The current title is “Synchrony of Bird Migration with Global Dispersal of Highly Pathogenic Avian Influenza Virus Clade 2.3.4.4 Reveals Exposed Bird Orders”.

- In the Introduction, the authors refer to the “re-emergence” of HPAIVs; in truth, I am not sure it is correct to talk about re-emergence considering that the virus has been circulating and evolving since 2008. It would be better to talk about resurgence or write the sentence as follows “The highly pathogenic avian influenza viruses (HPAIVs) subtype H5 clade 2.3.4.4 have caused unprecedented large numbers of wild bird deaths worldwide since 2014”

[Response] We have removed the use of “re-emergence” in the Introduction.

Reviewer #2 (Remarks to the Author):

This is an interesting paper on an important current topic, seeking to show how bird migration has explained the spread and incursions of the recent and ongoing highly pathogenic H5 2.3.4.4 outbreaks in different countries. The authors use data on bird migration from public sources (movebank), poultry trade data, and viral sequence data from GISAID in a global analyses covering at least 10 years. The paper is generally well written and the results look justified given the methods, so I have a few comments.

Minor and technical comments

1. You seem to use clade 2.3.2.1 as a comparator to 2.3.4.4 but dont really talk about this very much in the main text. Perhaps include a little in the introduction and discussion on this (a sentence or so?), e.g. is 2.3.2.1 is more 'poultry' driven than 2.3.4.4 ? (Figure 1B suggests not).

[Response] We have included additional text on clade 2.3.2.1 in the discussion (Line 266-277) and introduction (Line 57-61).

2. The sequence data is used to make time-scaled phylogenies, and discrete location and bird species traits are mapped onto the trees. I think you then use a binary network (line 257) in a discrete geographic region phylogenetic GLM framework but also include origin and destination sample size (non-binary) and shown in Figure 1B ?

[Response] The Reviewer is correct.

But did you (and why did you) use only one binary wild bird migration network (all birds species together) vs making a network for different species types ? I ask because the sea-birds / shore-birds might have a very different network than anseriformes (e.g. sea-birds / shore-birds go along the coasts and seas, not large distances inland).

[Response] We agree that some seabirds/shorebirds may have different migratory pathways compared to waterfowl. In fact, the pathways might be different within the same family, or even species. Nevertheless, we aim to conceptualize the migration in the current network where an edge represents if birds migrate between the start and end regions (according to currently available Movebank dataset) rather than the physical migratory trajectory. In doing so, even if seabirds and waterfowls may differ in trajectory, e.g., some seabirds migrate along the coast from South China to North China while some waterfowls migrate within the inland area, the link between the two regions applies to both types of birds.

And the binary network just connects the few discrete geographic broad regions, but what is the correlation between the bird migration network and distance between the regions, and so did you also try to include distance as a predictor ?

[Response] We calculated the distance between regions by great circle distance between the geographical coordinates of the points on the map (Figure 1C). Because the bird migration network is binary/categorical while the great circle distance is continuous, instead of calculating correlation, we categorize the distances into groups: 1) between locations where there is bird migration; 2) between locations without bird migration. We then tested if the mean great circle

distance of group 1 is significantly smaller than that of group 2, using unpaired two-sample t-test. While bird migration routes on average are shorter than average distance between all regions, they are not significantly shorter than average non-migration route distances (see the table below).

Mean distances of bird migration routes and non-migration routes					
	mean	standard deviation	t-test	t-test / t value	t-test / p value
great circle distances between regions	5927.43	3876.82	\	\	\
distances of bird migration routes (group 1)	4412.02	3605.89	smaller than great circle distances smaller than distances without bird migration (group 2)	-2.36 -0.78	0.01 0.22
distances of fall bird migration routes	4589.94	3688.69	smaller than great circle distances smaller than distances without bird fall migration	-1.52 -2.39	0.07 0.01
distances of spring bird migration routes	4234.10	3607.79	smaller than great circle distances smaller than distances without bird spring migration	-1.97 -2.91	0.03 0.00
distances between regions without bird migration (group 2)	4924.35	4292.27	greater than great circle distances	-2.25	0.99
distances between regions without fall bird migration	6690.71	3543.09	greater than distances with bird fall migration	1.79	0.04
distances between regions without spring bird migration	6743.04	3520.45	greater than distances with bird spring migration	1.92	0.03

We did not try to include great circle distance as a predictor because we aim to understand the underlying host movement process contributing to avian influenza virus lineage movements. Even if the great circle distance is predictive, it does not allow us to make any solid conclusions about the underlying host movement process. Some other distances, e.g., the distance of bird migration trajectory would be relevant predictors. In the future, with sufficient data, we expect to utilize bird migration trajectory data to construct the migration distance or traveling time network of bird migration. We have included this point in discussion of current study's limitation.

Also, the bird data is composed into presence and psuedo-absence on a grid (1km2), per species, is this used and aggregated to make the binary network? this seems a lot of work to get a binary network between large regions, but I notice that you use this data in a temporal phylodynamic analyses.

[Response] We would like to clarify that the binary network between large regions is constructed based on all publicly visualized studies (Table S9). Ideally, we would have used these data for the grids and further quantitative analyses. However, some of the data are not available to use in

research because of their data policy. Therefore, we used a subset of the data (Table S6) for the bird data in grids per species, which was used for further analyses.

3. Line 136 – if the sequences are highly clustered by region (large region), then why not use BEAST2/MASCOT or other structured coalescent framework, as this might help mitigate the biased rates results problems which are reportedly more evident in a discrete traits analysis? Note I'm not suggesting that you actually do this, because it might be that your datasets are too large at well over 1000 sequences each to get this to work in a sufficiently timely manner and that the results would not be sufficiently different anyway.

[Response] We agree that the structured coalescent framework offers an alternative approach to conduct similar analyses that are potentially less sensitive to sampling biases. Therefore, we have now included this suggestion in Discussion (Line 311-314) for future studies that would analyze smaller sequence datasets or fewer location states. We also agree with the major challenges to apply the structured coalescent framework in our study. While BEAST2/MASCOT should be able to work for 10-12 states (regions) in our case, the large number of sequences used make the approach computationally prohibitive.

4. Line 160 - did you try downsampling but maintaining diversity as much as possible? (e.g. X per bird species-type per month per country). Looks to me like clade 2.3.4.4 results in Figure 1B are affected by relatively over-sampled poultry? Or is this just because the poultry sampling is also reflective of the poultry population size? (or number of reported outbreaks because HPAI is notifiable).

[Response] We understand the reviewer's comment that "clade 2.3.4.4 results in Figure 1B are affected by relatively over-sampled poultry" refers to the fact that virus sample size at the origin location contributes to virus lineage movements. It is true that sampling bias might affect the result and samples in poultry might be relatively over-sampled. Unfortunately, the records of the host of virus samples are poorly standardized (Table S1). For example, for ~500 sequences with host "duck", it is unclear if they are domestic ducks or wild ducks. This level of data crudeness prevents us to properly down-sample or check if the poultry sampling is reflective of the population size or number of reported outbreaks as suggested.

For these reasons, we have undertaken a new analysis that controls for sample size at the origin location and directly compares the wild bird migration network and the poultry trade network. This analysis clearly supports the wild bird migration over live poultry trade for all three datasets (Table S4).

5. Lines 198, 208 - 216 - are these results affected by using only one binary bird movement network? (See point 2).

[Response] This is not case because the results in Section 2 are not based on the binary bird movement network. Instead, the results are based on the grids of bird distribution probability.

Also on line 202 - how many events are there from Europe -> Qinghai in your dataset? Are you speculating on the basis of essentially one event [branch] with under-sampled / unobserved in

between transmissions, how long is that branch (in time)? [I note in the text you put ‘we cannot conclude that’.. which I agree with].

[Response] The posterior mean number of Markov jump events from Europe to Qinghai in the inferred posterior trees of clade 2.3.4.4 is 0.42 (0-1, 95% HPDI).

We would like to clarify that we are not speculating based on one branch. We summarized the number of Markov jump events by counting the expected number of lineage movements (Markov jumps) between the locations in the posterior tree distributions (details in Line 456-462 in Supplementary), i.e., based on all branches that had an ancestor node inferred to be Europe and a leaf node inferred to be Qinghai in all posterior trees.

The distribution of time of all the events (midpoints of the branches) is shown in the figure below.

6. Line 382 Spelling - should be Markov Jumps (no e).

[Response] We have corrected the typo.

7. Figure S5 - Strong temporal signal in Tempest of HA

Are you sure you input a non-time scaled tree (e.g. maximum likelihood tree) into Tempest to generate these plots? Because it looks to me like you have included the plots using the BEAST MCC tree; since the root-to-tip distance (y-axis) is on the same time scale as the x-axis, which is not correct; you should use a non-time scaled tree for these plots.

[Response] We thank the reviewer for pointing out this mistake. We have updated the Tempest plots (Figure S5) by using non-time scaled maximum likelihood trees of clade 2.3.4.4 and clade 2.3.2.1.

Reviewer #3 (Remarks to the Author):

Highly pathogenic avian influenza virus H5 clade 2.3.4.4 has caused worldwide outbreaks in domestic poultry and increasing deaths of diverse species of wild birds since 2014, with massive outbreaks in novel wild bird species since 2021. In this context, the current study is proposing to address how seasonal wild bird migration facilitates global virus dispersal, and which avian species are exposed to HPAI H5 clade 2.3.4.4 viruses, with the idea of potentially shedding light on ongoing global outbreaks, with high relevance for conservation and global epidemiology. This is done by conducting phylodynamic analyses based on empirical data of bird movement tracking and virus genome sequences.

Although the topic is of great interest, a large amount of data was processed and the results appear to make sense, the study is disappointing because

(1) it does not put the results in context of the recent large scale outbreaks (for which much data is already available, notably in terms of wild bird species affected, but also of the virus clades involved, e.g., EFSA 2022),

[Response] We have addressed the Reviewer's comment by conducting analyses of more recent virus sequence data (2018 – 2023) using same methods (Line 371-375), and have updated our results (Page 4-6) and discussion (Line 278-282) accordingly.

(2) the presentation of some of the results make them difficult to understand (the whole part on the 'correlation between the virus lineage migration and the bird probability distribution at origin and destination regions' is confusing), and most importantly it suffers weaknesses due to the data used, notably on the avian ecology side of things. I concur with the authors that it is of importance to integrate wild bird ecology in avian influenza studies, but I think this is not done convincingly enough in the current manuscript.

[Response] We acknowledge that publicly available bird migration data are still limited; This is due to a long-standing problem in the bird tracking community that tracking data are often not made accessible for reuse in research – many data are either not archived or not standardized on a single unified platform, despite the many studies on this topic (also mentioned in Bernard et al. 2021). The bird migration data in this study are the most comprehensive data we were able to collect by reaching out to all researchers on Movebank. We think the current data are still reflective of the overall migration pattern by summarizing it as a high-level theoretical network (Figure S6); and the heterogeneity should be reflected by the location-specific probabilities of bird distribution we inferred from current bird tracking data by niche modeling. That being said, we agree that it is important to discuss the caveats due to data limitation. Therefore, we have extended our discussion on the limitations in integrating wild bird movement ecology in avian influenza studies (Line 283-299).

Bernard, A., Rodrigues, A.S.L., Cazalis, V. & Grémillet, D. 2021. Toward a global strategy for seabird tracking. *Conservation Biology* 14: e12804.

The 'Caveat' part (page 9) is welcome and a critical part of the manuscript, but the call for more data would gain to include some suggestions about sampling design issues. Also, the manuscript

would gain in discussing more clearly how the methods used may be affected by sampling limitation and how this might change with increasing availability of different types of data.

[Response] We thank the Reviewer for these useful suggestions. In the Discussion, we have elaborated on our previous suggestions on sampling designs, how the methods may be affected by sampling limitation and how this might change with increasing availability of different types of data (Line 300-321).

In the introduction (lines 83-86), it is stated 'Integrating host movement in studying HPAIV dispersal is important while challenging. One challenge is insufficient bird movement data, which causes that previous global-scale studies [23] cannot account for the high variation in bird behaviours across species and locations'. But does the current study do a good job at soundly considering variability in bird movement across species and locations? Using what can look as an impressive amount of data may not be sufficient to make reliable inference. The movement data used in the analyses is mostly the data freely available from Movebank, a repository where researchers declare and manage their animal movement data, some of them making their data available. Only a small fraction of the gathered data worldwide are freely accessible that way, and the data that was used, from looking at the content of the data set (and as described in the text), appears to be limited to only a very small and biased sample of the actual data acquired worldwide on bird movements. For instance, data from a total of 53 studies of all bird species was considered from Movebank in the current study (lines 308-309), leading to the use of movement data for a total of less than 4000 individuals, while in a recent paper reviewing tracking data only in seabirds, Bernard et al. (2021) say 'To optimize future tracking efforts, we performed a global assessment of seabird tracking data. We identified and mined 689 seabird tracking studies, reporting on > 28,000 individuals of 216 species from 17 families over the last four decades.' And of course seabirds only represent a small part of wild birds. A critical issue is thus what proportion of species were used in the current study compared to wild bird species or bird movement studies relevant for avian influenza circulation, and was the tracking data used in the analyses meaningful/biased? Bird (and animals/humans) do different types of movements and each of those may potentially have very different meaning for the risk of disease spread (e.g., Boulinier 2023). Many bird species are for instance central place foragers during breeding and this is when many are temporally equipped with GPS loggers to track those foraging movements; conversely, GLS loggers have been more used to explore large-scale migratory movements, which may be critical for some epidemiology questions, but they have much lower spatial precision). The data set used to characterize habitat use by bird orders is thus relatively small while very heterogeneous in terms of the types of considered movements (within a breeding season, across years...), which likely limits what can be interpreted from the results.

[Response] We consider this comment as one issue: was the tracking data meaningful or biased in terms of two aspects: 1) representative enough for various bird migration patterns; 2) relevant for avian influenza transmission or circulation?

Regarding the first aspect, we agree with the reviewer that the number of species is limited in our current study which is due to the low data availability for re-use in other research in the tracking

community (Bernard et al. 2021). Therefore, our current bird tracking dataset is the most detailed we could acquire.

We understand the Reviewer's concern on our data sufficiency when compared to the existing data. In response to the concern, we'd like to clarify the difference in existing data and availability for re-use in research. Although a large body of bird tracking data exist in many publications, as Bernard et al. 2021 pointed out, "...much of the existing tracking data are not yet available to other researchers and decision-makers in online databases". Therefore, the sample size of our current data is indeed small compared to all the existing data, but it is big as a reusable dataset.

Additionally, it is infeasible for us to reuse the existing data in publications. The challenges include that there is no data policy or guide, and that data are not standardized, etc. Those challenges have been recognized by the tracking community (Bernard et al. 2021). To make the existing data available to other researchers and conservation decision-makers, online databases like Movebank create standardized data storage, sharing and re-use frameworks.

As Bernard et al. 2021, and many papers they cite recommended: researchers and institutions should invest in sharing and archiving data in online databases and translate the value of tracking data into conservation. We think our study is a good (though inevitably limited) example of translating the value of bird tracking data into understanding on the risk of bird exposed to infectious disease and implications on conservation.

We also agree on the huge variability of the bird migration pattern and bird distribution and that our study is limited in addressing this issue within orders (Line 283-299).

Regarding the second aspect, first, we do not aim to address the related birds in avian influenza *transmission*. We aim to find birds that are more likely to be *exposed* to avian influenza by overlapping with the disease at the same time and at the same space and we define these orders "vulnerable/exposed orders". Transmission can occur when there is overlap or delay of overlap by environmental transmission. However, our study cannot assess the possibility of transmission as it involves more complex processes, e.g., host behaviors as the reviewer mentioned. In fact, currently, little is known about transmission mechanisms of HPAIV in wild birds. This is particularly true for inter-species transmission among seabirds, for which there is very few studies of the ecology of transmission. **We agree this should be on the agenda for future studies integrating host behavior and avian influenza transmission, but this is beyond the scope of our study.**

Nevertheless, our tracking data are indeed relevant for avian influenza *circulation*: the families that we use in tracking data (Table S6) include major wild bird species from which avian influenza viruses have been sampled (Table S1).

We have included more discussion about the data limitations and its relevance (Line 283-299, 326-336).

Regarding the first part on how seasonal bird migration associates with global HPAIV H5 dispersal, it is worrying that sample size is retained in the model as a key factor. As the authors stress, this could mean the patterns were affected by sampling bias, but it is unclear how the inclusion of sample size as a factor was sufficient to address potential biases, especially if there were (likely) interactions between sample sizes and spatial locations of more or less critical relevance for virus transmission.

[Response] We agree that simply including sample size in the model is not sufficient to address potential bias. To test if the sample size affects the relative importance of live poultry trade and wild bird migration, we have undertaken a new analysis that controls for sample size at the origin location and directly compares wild bird migration network and poultry trade network. The analysis clearly supports wild bird migration over live poultry trade for all three datasets (Table S4).

Also, when looking at the whole data set used on virus subtypes, it appears for instance that most data come from Anatids, while e.g. only 4 virus samples for clade 2.3.4.4 (over a total of 1845) come individuals of the order Sulliformes: how does this relate to key reported results about Sulliformes? And the same for other orders?

[Response] Virus sampling of avian influenza in wild birds is biased among species. It is due to many reasons including the varying visibility of the infected birds in passive surveillance and the varying accessibility of the bird capturing and virus sampling. Also, AIV genomic surveillance in wild birds is extremely difficult due to 1) incredibly low positive rates in randomly sampled healthy birds, e.g., in Zhang et al. 2023, 2) stringent protocols in capturing and sampling wild birds and 3) strict reporting procedures that are related to national obligations to international poultry trade treaties. Therefore, when we only summarise avian influenza related species from virus samples or outbreak reports, we may have ignored some other species that are less visible or accessible.

To reveal these potentially exposed species, we come up with our analysis framework. We utilize the virus genomic data to summarise the virus lineage movements. We then incorporate host movement data that include more species than only species that have been sampled of HPAIV. Therefore, by analyzing spatial-temporal synchrony of bird movements of different bird orders and virus lineage movements, we are able to detect bird orders that have overlap with the virus lineage movements, although they might not have virus samples sampled from them to date. Therefore, the detection of these potentially exposed bird orders in our results shows the possibility of more species that might have been exposed to avian influenza, though not studied or sampled to date.

Zhang, G., Li, B., Raghwani, J., Vrancken, B., Jia, R., Hill, S. C., Fournié, G., Cheng, Y., Yang, Q., Wang, Y., Wang, Z., Dong, L., Pybus, O. G., & Tian, H. (2023). Bidirectional Movement of Emerging H5N8 Avian Influenza Viruses Between Europe and Asia via Migratory Birds Since Early 2020. *Molecular Biology and Evolution*, 40(2)

Overall, more information should be provided on the potential sampling biases and their implications: what proportion of species of which bird orders were considered? What were the

feeding habits of the considered species within each orders? Raptors have common habits of feeding on other animals, notably warm-blooded ones, but within other bird orders the behaviours may be more heterogeneous, e.g., in some orders most species forage on food unlikely to be associated to exposure to the viruses, while others feeding on bird carrions and preys. This may affect within species transmission as well as among species transmission. Given the relatively small numbers of bird studies considered, this is an issue that needs to be addressed.

[Response] The Reviewer raises a good point. We address this by discussing the data biases in bird orders and their implications (Line 283-287). However, it should be noted that the host behavioral effect on virus transmission is beyond the scope of our study. As we have clarified in an earlier response, we do not aim to address the related birds in avian influenza *transmission*. We aim to find birds that are more likely to be exposed to avian influenza by overlapping with the disease at the same time and at the same space and we define these orders “vulnerable/exposed bird orders”.

Line 144: Live poultry trade was not included in terms of (net) fluxes of birds? How could that affect the results?

[Response] The live poultry trade was included in terms of trade value instead of net weights of traded poultry, because trade value data is the most complete. However, the trade value and the net weight (after removing 12% data without net weight) are highly correlated ($R^2=0.7973$, see the figure below). Therefore, we are confident that using trade data as a proxy does not affect the results.

Figure: the scatter plot of trade value (\$) and net weight (kg) of live poultry trade between countries at log scale. Source: United Nations Comtrade Database. Code and data are on GitHub repository.

Line 239: strange wording: ‘Historically, Anseriformes have been the focus of wild bird hosts when studying host-pathogen interaction in AIV studies’.

[Response] We changed the wording to ‘*Historically, studies of host-pathogen interaction of AIV have focused on Anseriformes as wild bird hosts.*’

Lines 299-301: yes, indeed.

Line 394: how is it accounted for that many lag times were considered?

[Response] We have noticed that previous statistical methods assumed a normal distribution of correlation coefficients which is not appropriate. To make the statistical method more sound, we have conducted a new analysis of the statistical association between virus lineage movements and bird migration. We then applied block bootstrapping to generate samples of monthly virus lineage movement counts from all events while keeping the year and the month of the event. The number of samples is decided by $20/0.05 \times \text{the number of routes} \times \text{the number of bird orders}$. We summarized the counts by month in each sample to get a monthly count sample. We calculated the correlation for each sample between the virus lineage migration of each route and each bird order distribution at the origin/destination region, respectively. To account for multiple comparisons of 9 bird orders, N routes, and 2 locations (origin/destination) on each route, we use p values $< 0.05/9/2/N$ to define the statistical significance in the correlations.

To focus on synchrony, we do not consider time lags between virus lineage movements and bird distribution. By doing so, we also avoid additional degrees of freedom. However, some bird orders that synchronize with virus lineage movements with time lags may not be detected in our current analyses. We have included this limitation in Discussion (Line 326-336).

In the reference list, some references are repeated (63, 64 ?)

It is strange that references to movement studies on Antarctic petrels are listed (48, 61, 104) while it is said in the Methods that studies on Procellariiformes were excluded (lines 314-318)(the relevance of Antarctic habitat is also unclear for the current study). Whether the data was eventually used or not (I understand not), it illustrates the strong heterogeneity of the considered data set on bird movements.

[Response] We have removed duplicate references (63 and 64). We also have removed the two Antarctic petrel studies (48, 61, 104) because Procellariiformes were indeed excluded due to their geographically restricted distribution.

Cited references

Boulinier, T. 2023. Avian influenza spread and seabird movements between colonies. Trends in Ecology & Evolution 38: 391-395.

Bernard, A., Rodrigues, A.S.L., Cazalis, V. & Grémillet, D. 2021. Toward a global strategy for seabird tracking. Conservation Biology 14: e12804.

REVIEWERS' COMMENTS

Reviewer #1 (Remarks to the Author):

The authors have carefully reviewed the paper. All the reviewers' suggestions have been properly considered and addressed at the best of their possibilities considering the well known gaps existing as regards the bird tracking data. The manuscript is now suitable for consideration for publication.

Reviewer #2 (Remarks to the Author):

Thank you for considering my and the other reviewers' previous comments in detail. You have made extensive revisions and introduced a new sequence set (now including 2018-2023 sequences from 2.3.4.4, but the 2.3.2.1 still only go to 2017), which has resulted in highlighting quite different detailed results, although the broadest scale conclusions remain the same.

Minor comments

Abstract

I agree with the idea to discuss both Clade 2.3.2.1 and 2.3.4.4, and not only focus on 2.3.4.4 as in the previous version. However, please check the wording of the abstract carefully, because just removing "2.3.4.4" makes it sound like high path H5 (any clade) has only been a problem since 2014, which is not the case. Suggest Lines 57-58: .."and increasing deaths of diverse species of wild birds, especially clade 2.3.4.4 since 2014" ?

Line 60 - no, it is not unclear in general how seasonal bird migration facilitates global virus dispersal, there are many previous studies on this, and you cite some of these yourself in the introduction (line 92-110). Suggest that you omit or re-word this sentence.

Line 62-65 - sentence OK, but add the clades that you are particularly studying at the end ?

Introduction

Line 78 - maybe start with the year "Since 2014, highly pathogenic avian influenza viruses.." ? [otherwise same problem as in the abstract]

Line 140 - I agree that it is reasonable to have early and recent clade 2.3.4.4 data sets based upon the evolutionary history of this clade. However, you should probably add a sentence here to explain why, and also whilst I realise that you do not want to get into the details of reassortments, it is likely worth mentioning here about early 2.3.4.4 being mostly H5N8 (are the H5N6 in your data too ?), whereas the most recent 2.3.4.4b are H5N1; and the 2.3.2.1 are H5N1 (different N1). But this also then raises the question of why you split at 2018, and not summer of 2019, 2020 or 2021 ? - it is far from clear from Figure 1A why splitting at 2018 was done, why not just do the upper clade separately from the lower clades ?

Results

Section 2.1 - make sure you distinguish between early and recent 2.3.4.4 data sets in the text.

Line 174 - 'become endemic' - what is the definition of endemic that you are using ? and do you mean in wild birds ? [suggest that you do mean in wild birds, and not in poultry] please clarify or omit.

Figure 2

Please reorder the legend so that the clades appear in the same vertical order as in the graph (same also for similar figures in supplementary)

Reviewer #2 (Remarks on code availability):

I have not re-run the code, but I have looked at the scripts and documentation, and it seems more or less OK. In the repo, the beast xmls are included (required to do the phylogeographic inference) and these contain all the parameters and sequence data needed. The authors should check the terms under which the sequence data itself may be re-distributed from GISAID, and may want to (i.e. probably should) include the data provider acknowledgement details in this repo as well as review the license under which they make the code and associated data available.

Reviewer #3 (Remarks to the Author):

The first version of the manuscript has been very thoroughly revised accounting for all comments and I now recommend its publication in Nature Communications. It is especially nice that the authors were able to up date the analyses with recently available data.

I only have minor remarks below, which could contribute to make the paper more valuable for the community of disease/animal ecologists:

Lines 266-268 : 'We assume that the bird population's birth and mortality are stable, and there is no external intervention. Therefore, the change in bird distribution probability is only due to immigration and emigration'. What do you mean by 'immigration' and 'emigration'? Classically in animal ecology, those are processes that link populations via the dispersal of individuals (sensus Clobert et al. 2001. Dispersal. Oxford University Press; i.e. from their birthplace to the place where they reproduce or from where they reproduced to where they reproduce). Immigrants are expected to settle (and reproduce) in a population. I understand you mean the flux of individuals via migration into an area or out of an area'. I strongly advise not to use the words immigration/emigration, but to use a wording that refers to the migratory movements into or from an area. I realize that etymologically, 'immigration' and 'emigration' may fit well, but it is confusing for population ecologists.

Line 268 : 'immigration and emigration' I suggest replacing by something like 'increase in local abundance due to a positive net flux of incoming individuals through migration, and decrease in local abundance due to a negative net flux of individuals via migration outside a given area'.

The same idea goes for Line 270 ('Emigration of Ciconiiformes', change to something like 'Southern migration if Ciconiiformes'), Line 273 ('Ciconiiformes might be exposed to clade 2.3.4.4 when they start emigrating from Europe', change to '...when they start migrating south from Europe'), Line 275, Line 285, Line 294, Line 298.

Also, on Line 337, change 'immigrate to Africa...' to 'have migrated south to Africa...'.

Line 368: insert 'were' or 'have been' to change 'that sampled avian influenza viruses' to 'that have been sampled for avian influenza viruses' or 'that were sampled for avian influenza viruses' (or do you want to stress that the species sampled the viruses?).

Line 375: you may want to insert 'including on movements other than migratory movements (Boulinier et al. 2016 or Boulinier 2023)' after 'With the increasing availability of bird tracking data,'. As I mentioned in my review of the previous version of the manuscript, movements other than migratory movements are often neglected, but may be critical for infectious agent transmission at some scales and some have been or should be the focus of more work; pointing that out may contribute to include more ecology at the interface between virology/epidemiology and avian biology (as rightfully stressed on lines 455-458).

(cited reference: Boulinier et al. 2016. Migration, prospecting, dispersal? What host movement matters for infectious agent circulation? Integr Comp Biol 56: 330-342.)

Reviewer #1 (Remarks to the Author):

The authors have carefully reviewed the paper. All the reviewers' suggestions have been properly considered and addressed at the best of their possibilities considering the well known gaps existing as regards the bird tracking data. The manuscript is now suitable for consideration for publication.

[Response] We appreciate the reviewer's positive assessment.

Reviewer #2 (Remarks to the Author):

Thank you for considering my and the other reviewers' previous comments in detail. You have made extensive revisions and introduced a new sequence set (now including 2018-2023 sequences from 2.3.4.4, but the 2.3.2.1 still only go to 2017), which has resulted in highlighting quite different detailed results, although the broadest scale conclusions remain the same.

Minor comments

Abstract

I agree with the idea to discuss both Clade 2.3.2.1 and 2.3.4.4, and not only focus on 2.3.4.4 as in the previous version. However, please check the wording of the abstract carefully, because just removing "2.3.4.4" makes it sound like high path H5 (any clade) has only been a problem since 2014, which is not the case. Suggest Lines 57-58: .."and increasing deaths of diverse species of wild birds, especially clade 2.3.4.4 since 2014" ?

[Response] We have revised the text accordingly.

Line 60 - no, it is not unclear in general how seasonal bird migration facilitates global virus dispersal, there are many previous studies on this, and you cite some of these yourself in the introduction (line 92-110). Suggest that you omit or re-word this sentence.

[Response] We agree that previous literature has acknowledged wild bird migration is an important ecological process contributing to the global dispersal of HPAIV H5. However, the gap remains at the global level in integrating bird tracking data of migration to quantify the mechanism and the synchrony. Therefore, we have re-worded the sentence to emphasize on these aspects as “However, the gap remains in integrating bird movement data of different species to quantify the contributing mechanism to the global virus dispersal. Addressing this gap also allows us to identify which avian species are exposed to the viruses and where.”.

Line 62-65 - sentence OK, but add the clades that you are particularly studying at the end ?

[Response] We have revised the text accordingly.

Introduction

*Line 78 - maybe start with the year "Since 2014, highly pathogenic avian influenza viruses.." ?
[otherwise same problem as in the abstract]*

[Response] We have revised the text accordingly.

Line 140 - I agree that is it reasonable to have early and recent clade 2.3.4.4 data sets based upon the evolutionary history of this clade. However, you should probably add a sentence here to explain why, and also whilst I realise that you do not want to get into the details of reassortments, it is likely worth mentioning here about early 2.3.4.4 being mostly H5N8 (are the H5N6 in your data too ?), whereas the most recent 2.3.4.4b are H5N1; and the 2.3.2.1 are H5N1 (different N1). But this also then raises the question of why you split at 2018, and not summer of 2019, 2020 or 2021 ? - it is far from clear from Figure 1A why splitting at 2018 was done, why not just do the upper clade separately from the lower clades ?

[Response] Regarding the split of clade 2.3.4.4 data, we have added “The split of clade 2.3.4.4 data is based on its history of evolution, epidemiology and sampling intensity, enabling us to

maintain the genetic diversity of the dataset while keeping the computational burden manageable.”.

We have also added (Line 112-115) that our analysis on HA would not be able to address the shift from N8 to N1 in clade 2.3.4.4: “We acknowledge two caveats. First, while we only included HA, the neuraminidase (NA) and internal genes contribute to virus evolution, e.g., via reassortment [24] and the major circulating subtype shift from H5N8 to H5N1 in clade 2.3.4.4.”.

Results

Section 2.1 - make sure you distinguish between early and recent 2.3.4.4 data sets in the text.

[Response] We have revised the text accordingly.

Line 174 - 'become endemic' - what is the definition of endemic that you are using ? and do you mean in wild birds ? [suggest that you do mean in wild birds, and not in poultry] please clarify or omit.

[Response] We have omitted the text.

Figure 2

Please reorder the legend so that the clades appear in the same vertical order as in the graph (same also for similar figures in supplementary)

[Response] We have reordered the legend in Figure 2 and S4.

Reviewer #2 (Remarks on code availability):

I have not re-run the code, but I have looked at the scripts and documentation, and it seems more or less OK. In the repo, the beast xmls are included (required to do the phylogeographic inference) and these contain all the parameters and sequence data needed. The authors should

check the terms under which the sequence data itself may be re-distributed from GISAID, and may want to (i.e. probably should) include the data provider acknowledgement details in this repo as well as review the license under which they make the code and associated data available.

[Response] We have included the data provider acknowledgement in the GitHub repository and reviewed the license.

Reviewer #3 (Remarks to the Author):

The first version of the manuscript has been very thoroughly revised accounting for all comments and I now recommend its publication in Nature Communications. It is especially nice that the authors were able to up date the analyses with recently available data.

I only have minor remarks below, which could contribute to make the paper more valuable for the community of disease/animal ecologists:

Lines 266-268 : ‘We assume that the bird population’s birth and mortality are stable, and there is no external intervention. Therefore, the change in bird distribution probability is only due to immigration and emigration’. What do you mean by ‘immigration’ and ‘emigration’? Classically in animal ecology, those are processes that link populations via the dispersal of individuals (sensus Clobert et al. 2001. Dispersal. Oxford University Press; i.e. from their birthplace to the place where they reproduce or from where they reproduced to where they reproduce). Immigrants are expected to settle (and reproduce) in a population. I understand you mean the flux of individuals via migration into an area or out of an area’. I strongly advise not to use the words immigration/emigration, but to use a wording that refers to the migratory movements into or from an area. I realize that etymologically, ‘immigration’ and ‘emigration’ may fit well, but it is confusing for population ecologists.

Line 268 : ‘immigration and emigration’ I suggest replacing by something like ‘increase in local abundance due to a positive net flux of incoming individuals through migration, and decrease in local abundance due to a negative net flux of individuals via migration outside a given area’.

The same idea goes for Line 270 (‘Emigration of Ciconiiformes’, change to something like ‘Southern migration of Ciconiiformes’), Line 273 (‘Ciconiiformes might be exposed to clade

2.3.4.4 *when they start emigrating from Europe*, change to *‘...when they start migrating south from Europe’*), Line 275, Line 285, Line 294, Line 298.

Also, on Line 337, change *‘immigrate to Africa...’* to *‘have migrated south to Africa...’*.

[Response] We appreciate the reviewer’s suggestion. We have revised the text accordingly. In addition, we have removed “immigration/emigration” in Table S4.

Line 368: insert ‘were’ or ‘have been’ to change ‘that sampled avian influenza viruses’ to ‘that have been sampled for avian influenza viruses’ or ‘that were sampled for avian influenza viruses’ (or do you want to stress that the species sampled the viruses?).

[Response] We have revised the text accordingly.

Line 375: you may want to insert ‘including on movements other than migratory movements (Boulinier et al. 2016 or Boulinier 2023)’ after ‘With the increasing availability of bird tracking data,’. As I mentioned in my review of the previous version of the manuscript, movements other than migratory movements are often neglected, but may be critical for infectious agent transmission at some scales and some have been or should be the focus of more work; pointing that out may contribute to include more ecology at the interface between virology/epidemiology and avian biology (as rightfully stressed on lines 455-458).

(cited reference: Boulinier et al. 2016. Migration, prospecting, dispersal? What host movement matters for infectious agent circulation? Integr Comp Biol 56: 330-342.)

[Response] We agree that for understanding finer-scale transmission, including movements other than migratory movements is important and currently lacking in literature. We added the suggested text and citation to Line 349-351 where we discussed about the inter-species transmission, as it may be more suitable: “With finer-scale bird tracking data on movements other than migratory movements [43], future studies could better integrate avian biology in understanding the inter-species transmission mechanism.”.